# Landscape features support natural pest control and farm income when pesticide application is reduced

Ana Klinnert [1] ✉, Ana Luisa Barbosa[1], Rui Catarino[2], Thomas Fellmann [1], Edoardo Baldoni [1], Caetano Beber [1], Jordan Hristov[1], Maria Luisa Paracchini[2], Carlo Rega[3], Franz Weiss[2], Peter Witzke [4,5] & Emilio Rodriguez-Cerezo[1]

Future trajectories of agricultural productivity need to incorporate environmental targets, including the reduction of pesticides use. Landscape features supporting natural pest control (LF-NPC) offer a nature-based solution that can serve as a partial substitute for synthetic pesticides, thereby supporting future productivity levels. Here, we introduce a novel approach to quantify the contribution of LF-NPC to agricultural yields and its associated economic value to crop production in a broad-scale context. Using the European Union as case study, we combine granular farm-level data, a spatially explicit map of LF-NPC potential, and a regional agro-economic supply and market model. The results reveal that farms located in areas characterized by higher LF-NPC potential experience lower productivity losses in a context of reduced synthetic pesticides use. Our analysis suggests that LF-NPC reduces yield gaps on average by four percentage points, and increases income by a similar magnitude. These results highlight the significance of LF-NPC for agricultural production and income, and provide a valuable reference point for farmers and policymakers aiming to successfully invest in landscape features to achieve pesticides reduction targets.

Sustainable use and management of biodiversity, including valuing, maintaining and enhancing ecosystem functions and services, is among the four long-term goals of the Kunming-Montreal Global Biodiversity Framework (GBF) adopted by the UN Convention on Biological Diversity[1]. Amongst others, the framework sets the 2030 global target of reducing the overall risk from pesticides and highly hazardous chemicals by at least 50%. Similarly, the European Green Deal promotes biodiversity preservation and restoration, and targets a reduction of pesticides use by 50% by 2030[2]. These goals are a direct response to mounting societal concerns and scientific evidence pointing at the negative environmental impacts of intensive agricultural systems[3–7]. The expansion of agricultural land for example,

and the intensive use of agrochemicals have resulted in simplified landscape structures and a loss of biodiversity in Europe and other regions of the world[8–12]. The loss of biodiversity has also led to a reduction in functional biodiversity, including natural enemies[11,13–15]. These organisms, including amongst others predatory insects, parasitoids, or birds are a valuable tool for sustainable pest management in agriculture, as they reduce pest populations, providing the ecosystem service of natural pest control (NPC)[16]. Consequently, natural enemies can reduce reliance on synthetic pesticides and in theory mitigate the suggested adverse impacts on agricultural productivity and its consequences resulting from the targeted 50% reduction in synthetic pesticides use[17–20].

[1]European Commission, Joint Research Centre, C/Inca Garcilaso 3, 41092 Seville, Spain. [2]European Commission, Joint Research Centre, Via Enrico Fermi 2749, 21027 Ispra, Italy. [3]European Commission, Directorate General for Agriculture and Rural Development, Brussels, Belgium. [4]EuroCARE Bonn GmbH, Buntspechtweg 22, D-53123 Bonn, Germany. [5]Institute for Food and Resource Economics, University of Bonn, Bonn, Germany. ✉e-mail: ana.klinnert@ec.europa.eu

In agricultural systems, NPC can be promoted through landscape complexity[21], which refers to small areas of natural or semi-natural vegetation within an agricultural landscape that can provide ecosystem services, promote biodiversity, and are often denoted as agricultural landscape features[22,23]. Agricultural landscape features can support the abundance of natural enemies, by providing shelter and alternative prey, thus allowing them to persist over time[24,25]. Accordingly, in scientific literature it has long been established that landscape complexity, with its associated diversity in flora and fauna, plays a crucial role in promoting NPC in agricultural systems and reduce reliance on chemical inputs[26]. Several recent empirical studies provide evidence of enhanced NPC resulting from higher landscape complexity[27–33].

While current research provides pivotal insights, it remains bottlenecked regarding a general quantification of the ecosystem service of NPC. Furthermore, links of NPC to crop yields, which is one of the main drivers of a farm's operability, are difficult to establish and often missing[34,35] due to multifaceted challenges. These challenges arise from insufficient data[36], inconsistent results[30,35,37–40], the underlying ecological complexity[41,42], or because results remain highly context-specific[37,43–45].

An additional challenge is the economic valuation of NPC as an ecosystem service. Losey & Vaughan (2006) estimate the monetary value of NPC in the United States at 4.5 billion USD per year, but acknowledge lack of data to further refine the evaluation[46]. Other studies have estimated the monetary value of NPC, but focus on a specific crop, pest, or geographic location, and hence do not provide a general economic evaluation[47–49]. However, especially under pesticides reduction targets, as outlined by the GBF and Green Deal, it becomes crucial to establish the economic value of NPC to crop production in more general terms. Policymakers and farmers must be aware of the financial benefits associated with landscape-driven NPC to undertake timely and appropriate investments in landscape management to meet the GBF and Green Deal targets[50] without encountering severe losses in agricultural production.

Taking the European Union (EU) as a case study, in this paper we provide a comprehensive assessment of the contribution of NPC potential driven by landscape features (hereafter referred to as LF-NPC potential), quantifying its impact on crop yields and the associated economic value in agricultural production under a reduced synthetic pesticides framework. For the assessment, we combine a European-wide dataset on yield observations derived from the EU Farm Accountancy Data Network (FADN) and a map of the continent's LF-NPC potential at 100-meter resolution. The LF-NPC map has been derived by merging high-resolution geospatial layers depicting landscape elements with extensive field surveys of flying insect predators and parasitoids[23]. The landscape elements considered were small woody features, grasslands, and forests, and their contribution to LF-NPC potential was determined based on observed flying insect abundance[23]. To further enhance the LF-NPC indicator, we combine it with a spatially explicit indicator on agricultural intensity across Europe[51] and a European crop mask[52]. This integration enables the computation of refined crop-specific LF-NPC scores at the regional level, providing a more granular representation of a region's LF-NPC status, particularly in relation to crop cultivation zones and areas with a higher concentration of organic farms (see Supplementary Information Section 1). The resulting indicator is the only one available which establishes LF-NPC potential across the European continent. At the same time, FADN provides farm-level yield observations and characteristics across the EU. Since synthetic pesticides are largely avoided in organic agriculture, we use the reported farm typology (organic or conventional) under FADN as a starting point to establish yield gaps capturing the difference in productivity between intensive (conventional) and less-intensive (organic) farming systems. To further narrow down the differences in yield gaps that can be attributed specifically to the avoidance of synthetic pesticides in organic farming,

we control for several influential factors that can affect yield differences between organic and conventional farms, such as purchased fertilizers and soil improvers. After estimating the regional yield gaps attributed to pesticides use for major crops, we investigate whether regions with higher LF-NPC potential exhibit a lower yield gap. Finally, to evaluate the economic implications of LF-NPC potential, we parametrize a partial equilibrium model and simulate the EU agricultural sector in 2030 taking into account a reduction in pesticides use and the previously estimated yield gaps, which are a function of a region's landscape complexity. By employing this approach, we aim to provide cutting-edge insights into the role of landscape design in shaping agricultural profitability in the context of reduced pesticides use.

## Results
The analysis included the following ten arable crops: wheat, barley, oats, peas, corn, potatoes, legumes, rye, durum wheat and fodder corn (see Method section for more details on crop selection). For these crops, we estimated yield gaps attributable to differences in pesticides use between organic and conventional farming practices. These yield gaps have then been compared to the regional LF-NPC potential.

### Yield gaps between organic and conventional farming
Using data from 2010 to 2017, yield gaps are estimated for each FADN region of the EU and by farm typology, provided that a minimum number of organic farms per farm typology and crop operate in a region. To ensure statistical robustness, FADN regions with fewer than 16 organic farms cultivating a specific crop were excluded from the analysis. Following this filtering process, a total of 534 regions and crop combinations remained, for which each time a uniform econometric model per crop, region and when possible, farm typology was employed. The models utilized the same farm-level variables as explanatory factors (see Methods for the selection of explanatory variables). This methodology resulted in 534 estimations of crop-specific regional yield gaps attributed to differences in synthetic pesticides usage. The regional, crop-specific econometric models have been provided with farm-level data points ranging from as few as 36 to as many as 20,222 for regions and crop categories with extensive observations, with each observation representing a distinct farm. The average regional yield gap in the EU between organic and conventional farming attributable to pesticides use from our estimations is −15%, with important differences associated to crops, ranging between −8% in legumes, −17% in barley and −23% in rye (see Fig. 1 and Supplementary Table 4). Regional differences are also substantial, for example varying between −6.5% (Norte e Centro−Portugal) to −18.7% (Baden-Wuerttemberg−Germany) in fodder corn.

### Contribution of LF-NPC potential to agricultural production
Correlating the LF-NPC potential of a region to the estimated crop-specific yield gap between organic and conventional farming due to pesticides use, shows a positive correlation for nine out of the ten analyzed crops (Fig. 2). The strength of correlation (Pearson) varied among crops, with values ranging from 0.14 for corn to 0.4 for potatoes. For crops with a large number of observations, such as barley, oat and wheat, the relationship was most of the times statistically significant (see Supplementary Table 3). Rye did not show any trend in the relationship between LF-NPC potential and yield gap.

Proceeding to the next phase, a mixed effect model was used to quantify the effect of LF-NPC on yield gaps. Results of the implemented mixed effect model were first obtained with a total of 326 observations including all ten crops and allowing both slopes and intercepts to vary across crop groups. The variation in the effect between crops is not very large, suggesting to drop the random component in the slope (see Supplementary Table 4). On the contrary, the random intercept variance in the model is significantly large, indicating a need to maintain a random effect in the intercept.

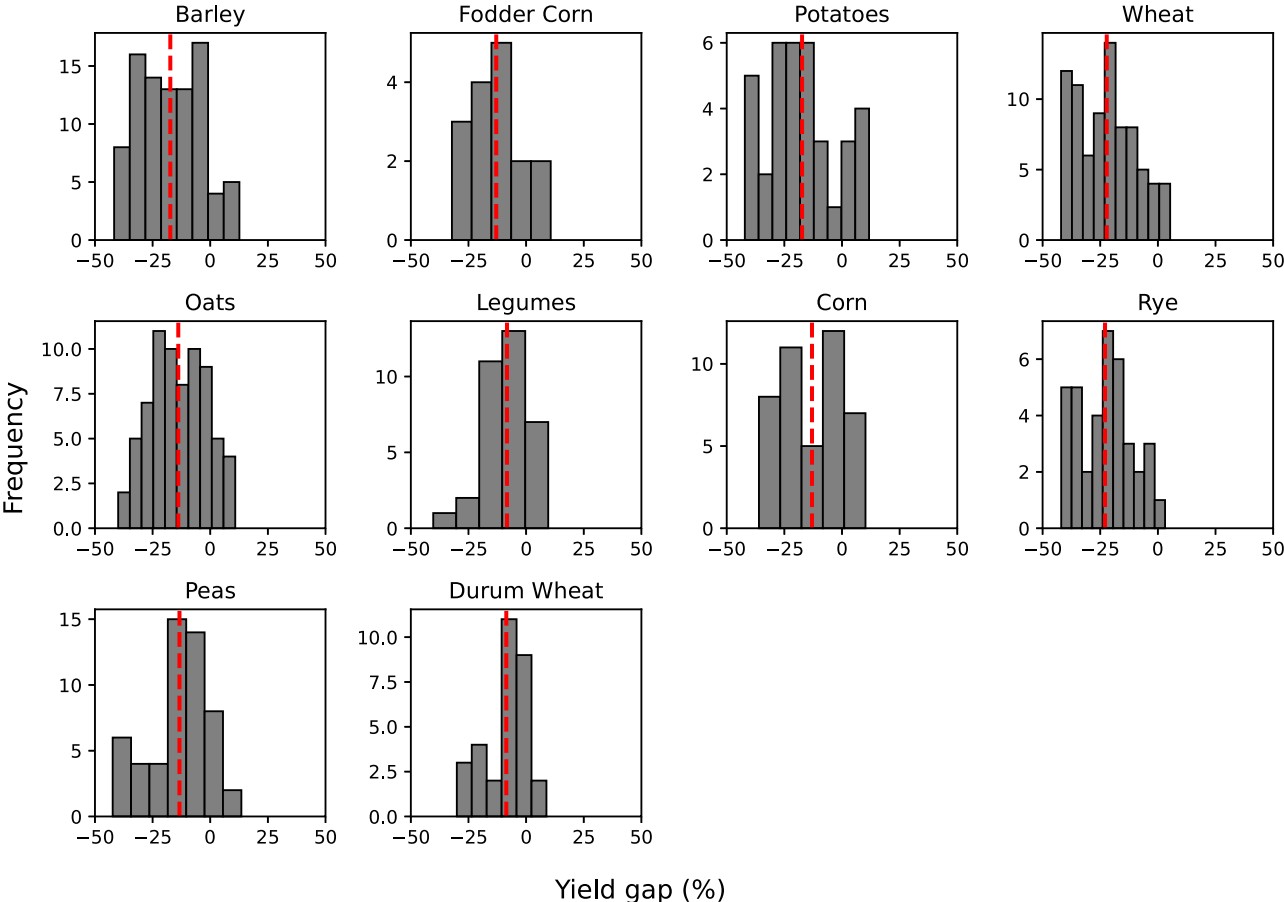

**Fig. 1 | Distribution of yield gap (%) estimation by crop.** Histograms show the distribution of regional (NUTS2 (Nomenclature of Territorial Units for Statistics)) yield gap estimates (%) per crop. These yield gaps are estimates of the productivity gap between conventional and organic farming due to differences in pesticides use. The dashed red line shows the median yield gap for each respective crop across all regional estimates. Source data are provided as a Source Data file.

Therefore, we use a mixed effect model, with only the intercept allowed to vary between crop groups, keeping the slope constant. The results indicate a statistically significant positive main effect of LF-NPC potential to yield gaps. Despite the positive coefficient, it is important to note that yield gaps are expressed as negative numbers. Therefore, an increase in a region's LF-NPC potential by one unit is associated with a 4.4 percentage point reduction in yield gap on average for all crops (Supplementary Table 4). To comprehensively interpret the significance of these findings, it is crucial to look at the variability and range of the LF-NPC scores across Europe. Generating crop-specific LF-NPC scores and normalizing by biogeographic zones, LF-NPC scores range from 0 to a maximum of 2.9 across Europe. The majority of regions cluster within the scores of 0 to 1.5, while the European average settles at 0.9. Notably, the European average suggests that between 2010 and 2017, LF-NPC accounted for a 4 percentage point (0.9 * 4.4%) reduction in yield gaps between organic and conventional farming practices, on average. Furthermore, our results suggest that regions characterized by the maximum LF-NPC experience a yield gap reduction of 12 percentage points (2.9 * 4.4%). The marginal R-square of 0.14, indicates that LF-NPC potential is able to explain 14% of the variability observed in regional crop yield gaps (see Supplementary Table 4). Using the established mixed model, regional yield gaps as a function of a region's LF-NPC potential are estimated across Europe.

**Quantification of the financial benefit of LF-NPC to agricultural production**

By incorporating these newly estimated, LF-NPC dependant regional yield gaps into the agro-economic, partial-equilibrium model CAPRI

(Common Agricultural Policy Regionalized Impact Modeling System), insights are gained regarding the financial benefit of LF-NPC to agricultural production. Specifically, we simulate a scenario in which all EU regions reduce their pesticides use and cost to levels observed in organic farming by 2030 and adopt yield gaps according to their LF-NPC profile (hereafter referred to as LF-NPC scenario). The results of the LF-NPC scenario are then compared to a business-as-usual scenario (baseline scenario), where neither yield changes, nor synthetic pesticides use reductions have been implemented. By comparing the income change between the baseline scenario and LF-NPC scenario between regions with varying LF-NPC potential, one can evaluate the financial benefits of LF-NPC in a reduced pesticide usage scenario, as the LF-NPC scenario considers factors such as the reduction in synthetic pesticide usage and cost, the LF-NPC potential of each region, and the corresponding crop-specific yield gap. This assessment is conducted holding all other factors constant, allowing for a focused analysis of the financial agricultural benefits associated with LF-NPC. Results show that, on average, a one-unit increase in the LF-NPC potential improves farm revenue by 3.7% and farm income by 4.2% per hectare (Fig. 3).

While on average a higher LF-NPC potential is linked to positive revenue and income changes, results also point at an important variability across regions with similar LF-NPC potential. This variability relates to differences in the agricultural structure of each region, such as the crop mixture grown, the associated yield gaps and the existing cost structure. To illustrate, one can consider two French regions, Picardie and Provence-Alpes-Cote d'Azur, which have similar LF-NPC scores. It is observed that the revenue in Picardie experiences a more

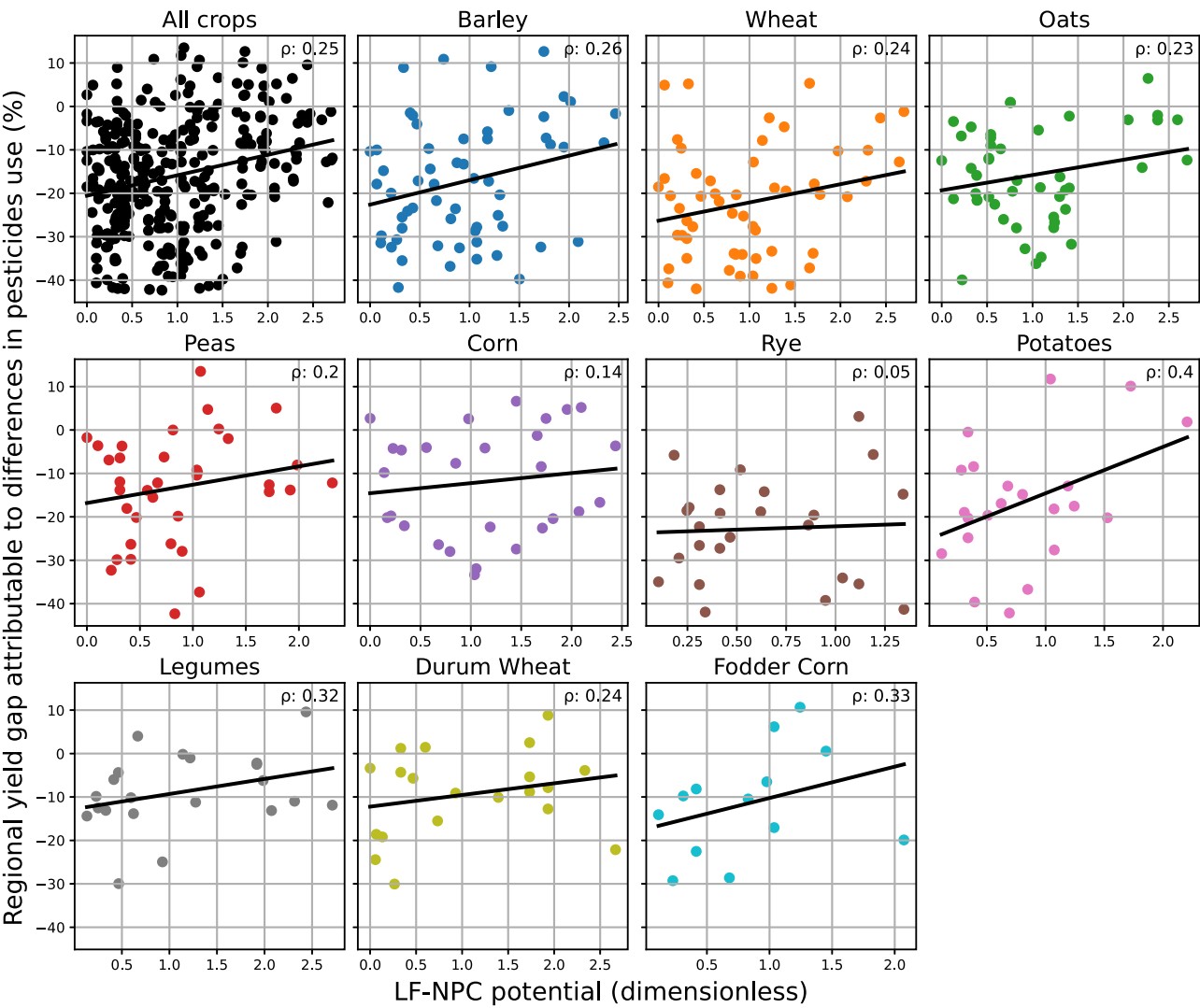

**Fig. 2 | Pearson correlation (ρ) between LF-NPC potential and yield gap (%) by crop.** Scatterplots show the relation between LF-NPC score and yield gaps. The black line in each chart represents the fitted regression line (linear). Each chart corresponds to a specific crop (see respective title), with the exception of the first chart (upper left corner), which shows the relationship between LF-NPC and yield gaps for all crops combined. Each point in each chart represents a region's (NUTS2) LF-NPC score (x-axis) and yield gap (y-axis). Sample sizes: All crops: $n = 326$; Barley: $n = 59$; Wheat: $n = 55$; Oats: $n = 47$; Peas: $n = 36$; Corn: $n = 28$; Rye: $n = 25$; Potatoes: $n = 22$; Legumes: $n = 21$; Durum Wheat: $n = 20$; Fodder Corn: $n = 13$. Source data are provided as a Source Data file.

substantial reduction compared to Provence-Alpes-Cote d'Azur. This is mainly attributed to the fact that Provence-Alpes-Cote d'Azur focuses primarily on growing fodder crops, which use and depend less on synthetic pesticides and thus the reduction in pesticides use has a smaller productivity and hence revenue impact. Conversely, Picardie grows mainly cereals, where the yield gap due to lower pesticides use is higher, compared to fodder crops, and hence, the reduction in pesticides use has a stronger negative impact on production and revenue levels. This being said, the reduction in pesticides cost due to the reduction in use is stronger in Picardie, nevertheless not sufficient to compensate for the loss in revenue. In some cases, when pesticides cost represents a very large share of the total cost, and the region's LF-NPC profile is very high hence the yield gap small, the revenue loss can be compensated by the cost reduction. The latter finding emphasises that the economic value of LF-NPC is especially high in regions where synthetic pesticides represent a large share of the total cost structure.

## Discussion
Recent global and EU targets for reducing pesticide use require an acceleration in the exploration of alternative pest control strategies

that can reduce reliance on chemical inputs. In our study, we investigate the economic advantages of incorporating landscape complexity to enhance LF-NPC potential and support agricultural productivity under a reduced pesticides scenario. Specifically, we compare the agricultural economic performance of regions with higher LF-NPC potential to those with lower potential. We estimated that a one-unit increase in LF-NPC potential, on average, results in a 4.4 percentage point increase in productivity and a similar increase in overall farm income.

In this study, LF-NPC serves as a proxy for potential abundance of species providing NPC[23] to sustain agricultural productivity when less synthetic pesticides are used. Therefore, the measured positive impact of LF-NPC can be interpreted as the beneficial feedback of natural enemies on farming productivity in the context of reduced synthetic pesticides use. Opposed to experimental field designs, the approach chosen is similar to the category of causal inference using a potential-outcome framework[53]: Observational data and domain assumptions are applied to build a statistical model connecting LF-NPC to agricultural productivity. Notably, our focus is on the direct benefits of LF-NPC potential for agricultural production and does not measure other

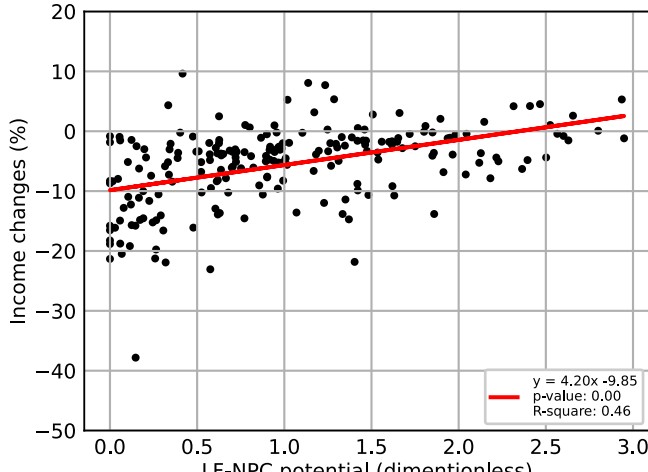

**Fig. 3 | LF-NPC potential versus income changes in percentage.** The scatterplot shows the relation between a region's LF-NPC score and its respective income change. The red line represents a simple linear fitted regression line. Each point represents a region's (NUTS2) LF-NPC score (x-axis) and the related income change (y-axis). The text box in the lower right corner shows the equation obtained from a simple linear regression of LF-NPC against income changes, the p-value of the coefficient (LF-NPC) using a two-sided paired t-test, and the adjusted R-square. Sample size: $n = 211$. Source data are provided as a Source Data file.

positive externalities of LF-NPC, such as soil erosion or carbon sequestration. The inclusion of these public goods would increase the total economic value of LF-NPC[54]. Additionally, our study does not undertake a comprehensive appraisal of the positive ramifications arising from a reduced use of synthetic pesticides, such as cost savings in water purification, health benefits for workers or nearby residents[55,56]. Furthermore, although we explore the potential positive feedback loop of an enhanced landscape design supporting NPC, we only accounted for the costs related to reduced pesticides use and did not explore the costs associated with redesigning the agricultural landscape to increase LF-NPC potential. Further investigation is needed to estimate these costs, which may involve leveraging diverse data sources.

Another critical aspect to consider is the utilization of flying invertebrates as a simplified model in this study to evaluate the potential of LF-NPC. The underlying data collection to construct the LF-NPC index focuses on flying invertebrates and not on the complete spectrum of natural enemies. For example, data on ground-dwelling organisms, known to significantly reduce insect pests and weed seeds in agroecosystems, were not collected[23,57–59]. Hence, our study represents one approach to assess the potential impact of LF-NPC, using flying invertebrates as a simplified model. This approach is grounded in the understanding that landscape complexity enhances the diversity of natural enemies beyond flying predators and parasitoids. Therefore, the analysis provides a broad-scale estimation that can be used for further research. Importantly, the findings fill the existing gap of a missing general quantification of the benefits of LF-NPC, which is essential for supporting further investments in redesigning agricultural landscapes, aligning with the biodiversity strategy for 2030 that promotes incorporating landscape features into agricultural land[60].

Previous studies have provided economic valuations of NPC in specific contexts, focusing on a specific location, crop or pest. For instance, one study valued NPC for soybean aphid management between 4.2 to 32.6 Euro per ha[61], while another found a 18% decrease in net farm income due to the loss of natural enemies in pear production[47]. Our study provides a general quantification of the regional benefits of LF-NPC potential across Europe under reduced pesticides use, considering multiple crops. The general quantification of LF-NPC potential allowed us to parametrize a European agro-economic partial equilibrium model in such way that we can examine the general economic value of LF-NPC potential for agricultural production. Taking into account regional agricultural sector characteristics, we obtain an estimate on the positive economic contribution of landscape features to farm income under reduced pesticides use. It is worth to note that when running a partial equilibrium scenario without considering trade feedbacks and related price effects from agri-food markets, lower yields generally result in decreased overall farm income. The main reason for this outcome is the decline in production levels while prices remain unchanged. When allowing for market and price feedback, higher prices due to lower production levels can potentially increase overall agricultural income compared to the business-as-usual (baseline) scenario (see Supplementary Information section 3). However, these price-related effects are not the focus of our analysis. The key insight here is the relative benefit of regions with higher LF-NPC potential compared to regions with lower LF-NPC potential in a reduced pesticides context. The financial benefit has been derived from the positive coefficient associated with LF-NPC potential in the regression analysis, when comparing a region's LF-NPC potential to future income changes (Fig. 3). The analysis also highlights the potential economic variability of outcomes given the same LF-NPC profile. Reducing synthetic pesticide use will have different financial impacts on regions, depending on their crop mix and cost structure. Regions growing crops which reap less benefits from LF-NPC and have relatively high yield declines due to reduced synthetic pesticides use, will experience more severe consequences. This variability should be taken into account in future research and policies aimed at optimizing pesticide reduction strategies across Europe.

Despite the observed variability in income, regions with a higher LF-NPC profile generally experience positive changes in income due to the established positive link between LF-NPC potential and yield gaps. Based on the FADN data, the average yield gap narrowed down to differences in pesticides use between organic and conventional farms and used to determine the contribution of LF-NPC-potential is −15%, with differences across crops and regions. Meta-analyses comparing overall organic and conventional yields often conclude that yields are highly contextual, depending on the system, site-specific characteristics and management practices[62–64]. These meta-analyses report average yield gaps between −19 and −25%, with considerable standard deviations[62–64]. Notably, organic fruits and oilseed crops show smaller gaps compared to conventional, whereas cereals and vegetables demonstrate larger gaps. Such differences might be related to the better organic performance of perennial over annual crops, and legumes over non-legumes. Our findings also confirm this, showing for example an average yield gap of −8% for legumes and −22% for wheat (see Supplementary Table SI 4). Additionally, field experiments that specifically assessed the effects of pesticides on yields also support our estimations. For example, Delin et al. found yield gaps ranging from −8% to −37% in wheat when comparing systems with and without pesticides use in Sweden, other production characteristics being equal[65]. Trials of Hossard et al. generated −5% to −13% wheat yield losses from a 50% pesticide reduction in France, corresponding to −24% to −33% in a zero-pesticide system[66]. While the estimated yield gaps in this study are consistent with those reported in previous research, the model structure and covariates used, might potentially underestimate the true yield gap. Specifically, if management practices that can help to bridge the productivity gap between organic and conventional farming are not fully captured, they might nuance the yield gap estimation and result in lower than actual yield gaps. In this study those management practices in some cases could only be approximated, such as the selection of high-yielding organic seed varieties or the implementation of effective crop rotation practices (see Method section).

Moving beyond the estimated yield gaps, we employed a mixed-effect model to quantify the potential impact of LF-NPC on reducing the estimated yield gaps. We found that higher LF-NPC profiles are positively linked to lower yield gaps, suggesting that landscapes providing more favorable habitats for beneficial insects not only enhance NPC but also reduce the existing productivity gap when less synthetic pesticides are applied. On average, a one-unit increase in LF-NPC leads to a 4.4 percentage point reduction in yield gap. With organic agriculture defined as a farming system that does not use synthetic pesticides or mineral fertilizers[67], reduced yield performance in organic farms can be attributed to nutrient limitations, pests and diseases. Pests and diseases play a more significant role in organic farming than in conventional farming, given the limited options for rapid nutrient replenishment or pest control through pesticides application[62,68]. Consequently, pest management in organic farming relies more on natural processes[63], such as the occurrence of natural enemies, whose abundance can be influenced by both management practices and landscape complexity[13]. Nevertheless, the susceptibility of different crop groups to insect pests may vary. It is, however, challenging to provide a precise quantification of crop-specific damage caused by insects across different temporal and spatial contexts. This challenge arises from various factors, including the diverse agricultural environments in which crops are grown and the dynamic nature of insect pests, which can undergo cycles of emergence and re-emergence[69]. An earlier study found that yield losses caused by insect pests range from 8 to 15% depending on the crop, with significant regional variations[70]. A more recent study analyzing yield losses of major crops caused by pests and pathogens confirms the significant role of several insect pests in modern agriculture, similarly highlighting substantial regional variations[69]. We accommodate for crop-specific susceptibility by allowing each crop to have an individual slope when relating yield gap to LF-NPC. However, this did not reveal substantial differences between crops. Additionally, interpreting the variation in correlation strength across crops between yield gaps and LF-NPC is challenging due to the lack of crop-specific quantifications of insect damage in organic farming. Despite this limitation, we can leverage the fundamental principle that insect pests have a natural predator or parasite and that organic farms encounter the same insect pests as conventional ones but rely on natural processes for their control[71]. Consequently, as insect pests remain key damage factors for crops[69], the positive correlation between yield gaps and LF-NPC aligns with the ecological rationale behind the potential of LF-NPC in mitigating severe pest infestations, thereby lowering the yield gap between organic and conventional farming.

The provided general quantification of LF-NPC's effect on agricultural productivity is crucial given the lack of a comprehensive understanding of its general trend and the varying outcomes of NPC at local level. Nevertheless, it is essential to acknowledge that while we have observed a higher LF-NPC potential linked to lower yield gaps, it does not guarantee effective pest control at local level. The effectiveness of NPC at local level can be influenced by various factors, including the severity of pest infestations, the balance between pest populations and natural enemies in the area, the effects of climate and climate change, or agricultural practices, such as crop diversity or field size[54,72-74]. Therefore, while our findings provide valuable insight into the general benefits of LF-NPC for reducing yield gaps, caution is warranted when interpreting these results as a guarantee of success at the local level[75].

Along this line of thought, it is important to also acknowledge that pesticide use intensity, reduced to organic farming levels in this study, can have considerable impact on biodiversity dynamics[76] and, consequently, can affect natural enemies[77]. However, within the CAPRI framework, there is currently no module that incorporates a coefficient to assess the impact of pesticide reduction on biodiversity, as finding such a universal measure is challenging due to its context-specific nature[45]. Ideally, future research could look towards employing process-explicit models that better account for the mechanisms influencing the distribution of organisms and communities, and how pesticides affect them at various scales. Integrating such modules that dynamically capture the complex interplay between pesticides, biodiversity, and NPC into the CAPRI analytical framework would significantly enhance its capabilities.

Finally, it is worth pointing out that, beyond LF-NPC, local diversification strategies like intercropping, cover crops, and small-scale[78], although context dependent, can offer farmers practical and timely solutions to pest management challenges[79]. These practices are known for their potential to enhance the presence and effectiveness of beneficial insects and other natural enemies in agricultural ecosystems[80], without necessarily taking land away from agricultural production. Therefore, such strategies can play a major role in reducing yield gaps in organic systems compared to conventional farming[64]. In summary, by analyzing the relationship between yield gaps and landscape design at a broader scale and at regional resolution, we overcome the limitations posed by local variability and identify a general trend of the benefits and value of LF-NPC for agricultural production. While the local level is ultimately crucial to realize the performance of LF-NPC, our results provide a so far missing reference point for farmers and policymakers on the potential of landscape features in contributing to agricultural productivity under pesticides reduction targets.

## Methods

The methodological approach is divided into three major parts. First, we estimate regional crop yield gaps between conventional and organic farming that can be attributed to differences in pesticides use. Second, we measure the contribution of LF-NPC potential to crop yield gaps, and third, we parametrize a partial equilibrium model to derive the economic value of LF-NPC potential.

### Input data

During the analysis two main high-resolution data sources were used:

1. A spatially explicit indicator of the LF-NPC potential at 100-meter resolution[23]. This indicator was derived by combining high resolution geospatial datasets on the presence and spatial arrangement of landscape features interspersed in the agricultural matrix with field level observations on the abundance of flying natural enemies associated to different landscape features. This layer was used in combination with a spatially explicit indicator on agricultural intensity across Europe[51] and a European crop mask[52] to enable the estimation of a refined crop-specific LF-NPC map. On the one hand, the indicator on agricultural intensity was used as a filter to focus on areas, which are likely to harbor a higher proportion of organic farms and hence be more relevant in terms of explaining the yield gap between organic and conventional farms. On the other hand, the crop mask was applied to restrict the calculation of regional LF-NPC scores to only LF-NPC pixels within the region, where the crop is cultivated (see Supplementary Information for a detailed description of the calculation steps). All three spatial layers are publicly available and have been retrieved from their publication location.

2. Farm level data collected under FADN. FADN surveys annually a large sample of agricultural holdings (around 80,000) across all EU Member States covering a wide range of farm types and sizes. Individual farm-level FADN data are not publicly available as they are covered by strict confidentiality rules under the GDPR regulation.

The data collected under FADN includes information on the physical characteristics of the farm, such as land use, yields or livestock numbers, farming management (organic or conventional) as well as economic and structural data, such as income and expenditure, assets

and liabilities, and subsidies received. As FADN data does not provide accurate geo locations of farms, observations of one farm cannot be directly linked to the geo-specific LF-NPC potential of that farm. However, FADN data does report the region (FADN region) within which a farm is located, and thus regional estimates of yield gaps can be estimated and paired to regional estimates of LF-NPC potential. Working at regional level due to the lack of information regarding the geolocation of farms, can limit our understanding of the influence of context-specific factors explaining the relationship between LF-NPC potential and yield gaps[75]. However, it also enables us to mitigate the variability caused by unique circumstances and identify more widespread patterns that are applicable across Europe.

### Yield gaps between organic and conventional farming
Before performing the yield gap estimations, we carry out data cleaning based on the examination of yield data. For each Member State, we analyze the yield distribution for each crop. The data cleaning is performed by removing the extreme 1% from both tails of each distribution, i.e., the most extreme 2% of the observations are removed. These data points are identified as outliers and often deviate from biophysical plausible yield values. Using FADN data, we confine the estimation of yield gaps to each FADN region and, whenever possible, to each farm typology to ensure both representative data and limiting the heterogeneity of the operating environment of farms. In regions with over 16 reported organic farms per farm typology and crop, we calculate within each region yield gaps for each farm typology and crop. These crop-specific yield gap estimates per typology are then averaged regionally to obtain a single estimate for each crop in that region. In regions with insufficient organic observations to segment the analysis by farm typology, we derive a regional yield gap by aggregating different farm types together. Additionally, crop-region combinations with fewer than 16 organic farms across all farm typologies were excluded entirely from the analysis. Using this approach, 77% of yield gaps were derived from farm typology-specific estimations, while 23% were obtained from estimations across farm typologies.

Furthermore, as neither quantity nor type of applied pesticide are available in FADN, and since synthetic pesticides are largely avoided in organic agriculture, we make use of the organic status reported for each farm to analyze the potential impact of variations in pesticide use on crop yields. For each selected crop $k$ in each region $i$, we first define the (log) crop-yields ($y_{ft}$) for farm $f$ at time $t$, using a linear multiple regression to control through $\mathbf{x}_{ft}$ for structural and geographical characteristics of the farm, farming practices, as well as year-specific shocks, such that the coefficient ($\beta_2$) of the fully organic status variable ($organic_{ft}$) is narrowed down to capture yield differences that are attributable to the use of pesticides. $\epsilon_{ft}$ represents the spherical disturbances.

$$\log\left(y_{ft}\right) = \beta_0 + \boldsymbol{\beta}_1' \mathbf{x}_{ft} + \beta_2 organic_{ft} + \epsilon_{ft} \tag{1}$$

$$\epsilon_{ft} \sim N(0, \sigma_\epsilon^2)$$

Then we estimate the yield gap ($\Delta y_{k,i}$) for crop $k$ and region $i$ between organic and conventional farming attributable to differences in pesticides use in percentage as:

$$\% \Delta y_{k,i} = \left(e^{\widehat{\boldsymbol{\beta}_2}} - 1\right) * 100 \tag{2}$$

To ensure that we capture the effect of year-specific shocks that may influence yields in a local area, we include yearly time dummies in $\mathbf{x}_{ft}$. Included in this matrix of covariates is also a dummy variable representing partially organic farms (i.e. growing both organic and

non-organic crops) and those that are in the process of conversion. To better control for factors such as soil quality, water availability, natural constraints and other local factors[63,81,82], we also include sub-regional (NUTS3), and altitude dummies. Furthermore, to control for structural characteristics and farming practices we include specialization dummies, information on the irrigation system, the share of unpaid labor, ratio of subsidies to gross income, the share of rented land, the ratio of assets to liabilities, assets per hectare, the capital-labor ratio, the physical size of the farm in terms of utilized agricultural area (UAA), and the share of revenues coming from crop activities. All these variables are aimed to control for the technological and professional nature of the farm as well as the possible presence of economies of scale which may have an influence on its performance. Moreover, to exclude the effect of fertilizers on yield gaps, we also control for purchased fertilizers and soil improvers. As the latter variable excludes manure produced on the holding, which can represent a significant source of fertilizers, we further add a covariate capturing the number of livestock units divided by the hectares of UAA, to proxy manure produced on the farm. Total livestock units are the sum of the number of equines, cattle, sheep, goats, pigs, and poultry (annual average number), converted into a standardized measure, i.e., livestock units. Livestock units are determined by multiplying each animal category by a conversion coefficient representing the nutritional or feed requirement of each type of animal[83]. The reference unit used for the calculation of livestock units (=1 LSU) is the grazing equivalent of one adult dairy cow producing 3000 kg of milk annually, without additional concentrated feedstuffs. As feed requirements are correlated with manure production, the total number of livestock units per hectare proxies the amount of available manure for use on the holding that could be used as fertilizer. In addition, to account for further differences in management practices between organic and conventional farm, we include seeds costs and a proxy for crop rotation. In the absence of qualitative information regarding seeds quality and their pest resistance, we assume that the use of varieties less sensitive to pest is correlated with the price of seeds, an assumption in line with findings from the literature[84–86]. As the same farms are not reported in consecutive year within the FADN database, crop rotations are captured in a simplified way through estimating the Shannon Index on shares of cultivated hectares by crop. Therefore, the implemented Shannon Index captures crop diversity, which has been occasionally used as a proxy of crop rotation, especially in simulation studies based on static positive mathematical programming using FADN data[87–89]. Finally, by focusing on yield gaps rather than solely organic yields, we also account for variations in yield standards, such as climate, that is not reported in the FADN database. The assumption is that after controlling for all these factors the obtained yield gap is attributable to differences in pesticide use. In other words, the organic status coefficient ($\beta_2$) is expected to capture the change in yield related to pesticides use between organic and conventional farming.

### Contribution of LF-NPC potential to agricultural production
The contribution of LF-NPC to agricultural crop production was conducted for the following ten arable crops: wheat, barley, oats, peas, corn, potatoes, legumes, rye, durum wheat and fodder corn (Supplementary Information Section 2 provides more details on crop selection). For these crops, we estimated yield gaps attributable to differences in pesticides use between organic and conventional farming practices. These yield gaps have then been compared to the regional LF-NPC potential.

By confronting the obtained regional yield gaps to a region's LF-NPC potential, we investigate across Europe, whether regions with higher LF-NPC potential face a lower yield gap due to the reinforced presence of beneficial insects (Fig. 4). To obtain the median LF-NPC potential of a region, we compute zonal statistics[90] of each region in Europe based on the 100-meter resolution map of LF-NPC elaborated

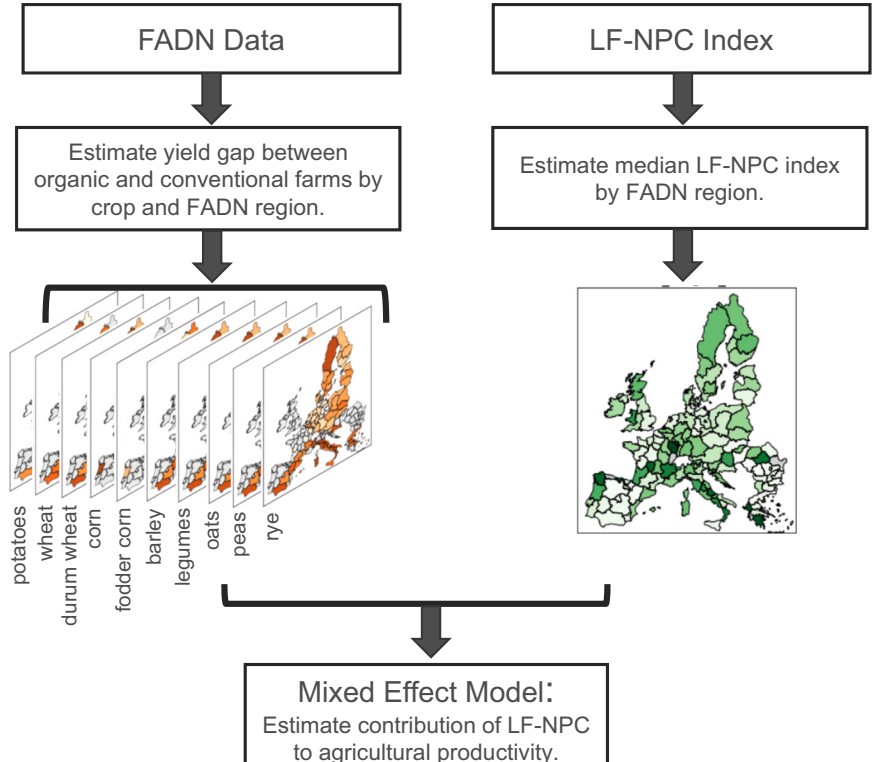

**Fig. 4 | Schematic workflow to estimate the contribution of LF-NPC potential to agricultural production.** Workflow representation showing the two main data sources (FADN and the LF-NPC Index) used in the mixed effect model. The mixed effect model is used to evaluate the relationship between LF-NPC and yield gaps. While FADN is used in regional regression models to obtain crop-specific yield gaps between organic and conventional farming due to differences in pesticides use, the LF-NPC Index is used to derive a region's (FADN region) median LF-NPC score.

by Rega et al. Additionally, to further account for natural conditions, which are required to describe the effective capacity of the ecosystem to deliver the service[91], we standardized the index based on each region's biogeographical classification (see Supplementary Information Section 1)[92]. Further filtering is conducted by focusing on areas of low and medium agricultural intensity, which harbor a higher share of organic farms. A final step involved computing crop-specific LF-NPC scores using a 10 m crop mask as baseline. The generated regional LF-NPC indexes are then paired with the obtained regional yield gaps.

To quantify the contribution of a region's median LF-NPC potential to the estimated yield gap, we employ a mixed effect model. Linear mixed models are often used when there is a nested structure to the data, meaning that the relationship between response and explanatory variable can be grouped into different classes[93]. In our model, we group the relation between LF-NPC potential and yield gap into different crop categories, allowing each crop group to have its individual relation (slope and intercept) to LF-NPC potential, while accounting for observations across crop groups. Crop category therefore becomes the random effect. Estimates of the model's coefficients capture the change in yield gap attributable to a one-unit increase of LF-NPC potential. The model is specified as:

$$y_i = NPC_i \cdot \beta + NPC_i \cdot b_i + \varepsilon_i \qquad (3)$$

$$b_i \sim N(0, \Psi)$$

$$\varepsilon_i \sim N(0, \sigma^2 I)$$

where $y_i$ is a vector representing yield gaps between conventional and organic farms, $NPC_i$ a vector of known regional LF-NPC potential values, $\beta$ a vector of fixed effect coefficients, $b_i$ a vector of random

effects coefficients and $\varepsilon_i$ the error term representing the unaccounted variability in the observations. The random effect vector $b_i$ and the error term $\varepsilon_i$ follow multivariate normal distributions with $\Psi$ as the covariance matrix for the random effects and $\sigma^2 I$ representing the covariance matrix for the error component[94]. Furthermore, we calculate marginal and conditional $R^2$ values to analyze the variation explained by fixed versus random effects[95].

Two advantages of employing a mixed-effect model in our analysis can be highlighted. Firstly, while the relationship between LF-NPC potential and yield gap may vary between crops, a higher LF-NPC potential is expected to generally reduce the yield gap. Therefore, observations in one crop group should not be modeled completely independently, as would be the case in a simple linear regression conducted by crop. The mixed-effect model allows us to model all crop groups together, while accounting for each crop group's individual relation to LF-NPC potential, thus providing a more accurate representation of the data. Secondly, as we are working at this stage at regional level, fewer observations are available for each crop (not all considered crops are present in all regions). By pooling observations across crop groups in a mixed-effect model, we can compensate for the reduced sample size at the regional level. Thus, this approach ensures that the study has sufficient statistical power to accurately estimate the effect of LF-NPC potential on crop yields across Europe. Therefore, by using a mixed-effect model, we can address the potential confounding effects of crop type and regional variation in LF-NPC potential on yield gaps, providing a more comprehensive understanding of the contribution of LF-NPC potential to agricultural production.

**Parametrization of the agro-economic partial equilibrium model**
To evaluate the economic benefit of LF-NPC under reduced pesticides conditions, we parametrize the CAPRI modeling system. CAPRI

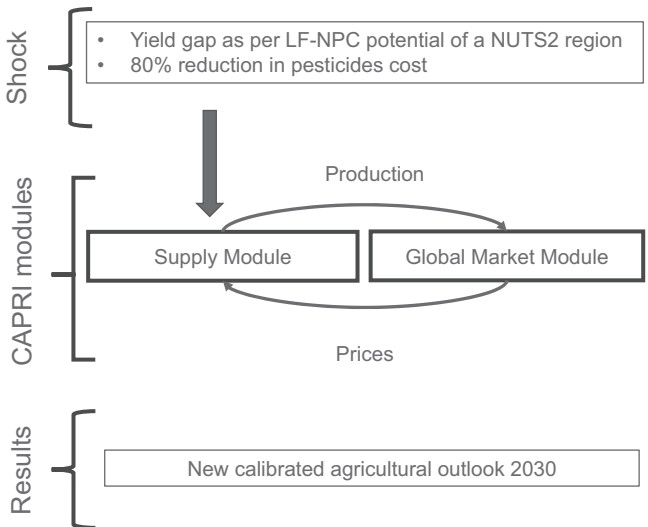

**Fig. 5 | Partial equilibrium model with description of changes inserted into the supply module.** Yield gaps corresponding to a region's LF-NPC index are incorporated into the partial equilibrium model CAPRI, alongside a substantial reduction in pesticide costs. These adjustments lead to modifications within the supply module, forcing the model to seek a new equilibrium that reconciles agricultural demand and supply. The simulation results in a recalibrated agricultural outlook that reflects agricultural and economic consequences of the reduced pesticide usage when taking regional LF-NPC potentials into account.

is a global, comparative static, partial equilibrium model for the agriculture and the primary processing sectors. CAPRI has a highly detailed and disaggregated representation of regional agricultural production and supply within the EU. Specifically, the CAPRI EU supply module comprises about 280 independent optimization models that represent regional agricultural production activities at the EU NUTS2 level. These optimization models consider intermediate inputs for production activities, incorporate non-linear cost functions, and include constraints related to land availability, nutrient balances, and policy restrictions at the regional level. The EU supply models are interlinked with an international market model via an iterative process: commodity prices from the global market module enter the profit maximization system of the EU regions, while EU agricultural supply from the regions is fed into the market module, which in turn re-calculates the market balance and commodity prices. CAPRI is frequently used for impact assessments of agricultural, environmental, climate change and trade related issues, both at EU[96–98] and global level[99,100]. In this study, we project all EU regions to reduce their pesticides use to levels observed under organic farming by 2030, and adopt the estimated yield gaps, taking into account each region's LF-NPC potential and the pesticides reduction. The latter is implemented by changing yields and pesticides cost under the regional agricultural supply modules of CAPRI (Fig. 5). The yield gaps, a function of a region's LF-NPC, are introduced as exogenous shocks to the model. Similarly, pesticides cost are forced upon the model exogenously. In this simulation, pesticides costs are reduced by 80% in the model, assuming that although synthetic pesticides use is reduced to zero, organic farmers still incur some crop protection costs. The market module is activated in an additional scenario to allow that changes in trade and international commodity prices enter the profit maximization of the supply module (see Supplementary Information Section 3).

## Reporting summary
Further information on research design is available in the Nature Portfolio Reporting Summary linked to this article.

## Data availability

FADN: https://agriculture.ec.europa.eu/data-and-analysis/farm-structures-and-economics/fadn_en. The individual FADN data are protected and are not available due to data privacy laws: "According to Regulation (EC) No 1217/2009 the data are covered by strict confidentiality rules and can only be used to meet the needs of the common agricultural policy. For example, they cannot be used by authorities for tax or compliance purposes." LF-NPC as per Rega et al. 2018 is available at: https://www.sciencedirect.com/science/article/pii/S1470160X18302309. European crop mask 2018 is available at: https://www.sciencedirect.com/science/article/pii/S0034425721004284. Classification of European agricultural land crop-management systems is available at: https://www.sciencedirect.com/science/article/pii/S0169204618314440. Source Data: All data generated in this study are provided in the Source data file provided with this paper and are accessible through the GitHub repository: https://github.com/anakl/LF-NPC. Source data are provided with this paper.

## Code availability

All analysis code and output are available through our GitHub project site https://github.com/anakl/LF-NPC and are released on Zenodo (https://doi.org/10.5281/zenodo.11040631)[101]. Data analysis and models are run using R (version 4.2.2) and the CAPRI model (TRUNK version−October 2022).

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

## Acknowledgements

We would like to thank Kevin Schneider for his insightful discussions and valuable input as well as Raphael d'Andrimont for providing expertise on the European crop layer used in this analysis. No funding was received to develop this study.

## Author contributions

A.K., A.L.B., and R.C. designed the research framework with M.L.P., E.R.C, F.W. and C.R. providing their expertise in natural pest control and sustainable farm practices. E.B. and C.B. performed the yield gap analysis. A.K. and C.R. worked on the mixed effect model. A.K., J.H and P.W. carried out the technical implementation and scenario simulation in the agro-economic model. A.K. and E.B. produced the figures. A.K and T.F. led the writing of the paper. All authors provided feedback, and contributed to the interpretation of the results.

## Competing interests

The authors declare no competing interests. The views expressed are purely those of the authors and may not in any circumstances be regarded as stating an official position of the European Commission.
