## [Peer Review File · Nature Communications]

Reviewers' Comments:

Reviewer #1:

Remarks to the Author:

This is a well written and interesting paper mapping the magnitude of pest control across Europe and relating it to yield gaps if synthetic insecticides are avoided. Even though I find the approach commendable, I have several doubts about the pest control and yield gap estimates. To make these estimates more convincing I would have liked to see validation with smaller datasets using real estimates of pest control (e.g. from sentinel prey cards and exclusion cages) and through real estimates of yield gaps due to pesticides when keeping everything else constant.

Major comments:

1) I have doubts about the validity of the calculations of the study for three main reasons. First, the study estimates yield gaps attributable to synthetic pesticides by comparing yields in organic vs conventional fields. Purchase of mineral fertilizers and a number of other factors are accounted for and remaining differences in yields are assumed to be due to pesticide use. In my mind this still remains a fairly shaky assumption, since there are still lots of differences between farming systems not accounted for. Organic fields often use non-synthetic pesticides, may use varieties less sensitive to pests, use different crop rotations which may have strong effects on pests and enemies. Furthermore organic farms are often located in more extensive agricultural areas, and these differences may not be captured when averaging across regions. Second, the pest control estimates use flying invertebrate predators and parasitoids as a proxy. This approach has several limitations. It ignores ground dwelling predators which in some systems are key natural enemies, and it ignores that pests respond very differently to landscape context (see e.g. Karp, D.S. et al. 2018. Crop pests and predators exhibit inconsistent responses to surrounding landscape composition. PNAS, 115 (33) E7863-E7870. <http://www.pnas.org/content/115/33/E7863>) Third, the mapping is done across several different crops without adjusting to the relevant crop pests. Since pests respond differently to landscape composition and are accessible to different groups of natural enemies, flying natural enemies may not be a strong proxy for pest control across agricultural crops.

2) Lines 124-129. These statements seem contradictory: 'The results indicate a statistically significant positive main effect of LF-NPC potential to yield gaps. Increasing a region's LF-NPC potential by one unit results in a 6.2% reduction in yield gap on average for all crops'. The first statement is supported by fig 2. The second statement sounds like the opposite pattern? According to the figures an increasing LF-NPC seems to result in an increase in the yield gap. Same interpretation on line 14-15. Is there something wrong here or am I missing something important?

Minor comments:

- 3) Line 36-39. Not only seminatural habitats contribute to landscape complexity. Also crop diversity and field size can contribute strongly and both have been shown to be quite important for natural pest control.
- 4) Line 67. Do you mean 'flying insect predators and parasitoids'?
- 5) Line 183, 185 and 189. Why 'species richness'?
- 6) Figure SI 3. This is a table, not a figure.
- 7) Supplementary material line 58-59. This sentence is unclear. Something missing at the end?

Reviewer #2:

Remarks to the Author:

This study attempts to estimate crop yield and economic outcomes of reducing pesticide use in Europe, taking into account the fact that the natural pest control is likely to depend on the landscape features. An ambitious and timely approach given the environmental commitments at international, EU and often also national levels. The "trick" the authors are using is assuming that organic yields (controlling for synthetic fertilizer use) represent the yield levels under conventional agriculture with reduced pesticide use. Yield gaps [organic-conventional] are calculated and regressed against an index for landscape-features supporting natural pest control (LF-NPC) at the

level of the FADN regions (FADN is a state-of-the art farm panel dataset), controlling for other farm-level characteristics. The results suggest that the yield gap is reduced as LF-NPC increases. This is as far as I know a novel and very interesting result in itself, even though I lack a discussion of why LF-NPC should be a good predictor for yield (gaps), and a more critical approach when it comes to the modelling (see comments below). In a second step, to simulate what would happen under a drastic reduction of pesticide use the authors introduce the yield gaps and reductions in pesticide costs into the CAPRI model, a state-of-the art European agricultural sector model. They find that the income changes are also reduced – incidentally by almost exactly the same amount - if there is a higher LF-NPC in the (FADN) region, partly through the yield gap effect, and partly through the difference in pesticide use between regions and crops.

A very interesting study overall, yet I have a number of concerns that I think need to be addressed before this study can be reassessed.

* on the level of detail in terms of methods and results

I would like to see much more detail on results, model diagnostics, removal of outliers, which regions were or were not included in the end. This is important to assess the validity of the approach. See comments below for more specifics.

* On using organic as pseudo-treatment for conventional with no (or much less) pesticides

Organic farming often is different from conventional farming in many ways. While covariates have been used for the estimation of yield gaps to reduce that problem, issues remain. For example, organic farmers are often maintaining soil fertility and suppressing weeds by having longer rotations (that often include grass-clover leys) than conventional farmers – unclear if this is taken into account in the CAPRI part of the study, I think it is just the yields that are changed not the rotations. I would like to see a version of the CAPRI modelling where rotations are also adapted. I think this could give a range of what we would expect in future if pesticide use is reduced significantly.

* On the LF-NPC data and its use and communication:

There is little discussion about the ecological motivation for using LF-NPC for explaining yield gaps, even though Rega et al. 2018 are very careful about discussing limitations in interpretation and onward use. Is there other evidence suggesting that yield gaps between conventional and organic are due to pests/pesticide use? Weeds and fungal diseases may be a more important limiting factor and are not really covered by LF-NPC I think.

I would recommend computing a crop-specific LF-NPC estimate for each region using the freely available EU crop type map for 2018. Indeed some of the crops may be grown in parts of the FADN region only, thus biasing the LF-NPC estimates for a particular crop away from their true value?

The heavy referring to Rega et al. 2018 makes understanding and communication of the results more complicated than necessary. In the Abstract, the statement “a one-unit of LF-NPC potential, on average, leads to a 6.1% increase in agricultural income” is meaningless if the reader does not get an idea of what that unit represents. I guess this can be different things, but in that case replacing by an example would be better. If length is a concern, cut down a bit on the linear mixed models details, with which many readers will be familiar). This is compounded by the fact that it is not actually LF-NPC that is used but the LF-NPC standardized by biogeographic region. This is not well detailed anywhere, not even in the code.

L328 L330 the standardization of the LF-NPC needs to be explained more in detail. Why is it done and does the use of unstandardized data give very different results?

* On the FADN data:

Motivation for the use of FADN regions: I would like to see some evidence for the variability within and between FADN regions of the yields and LF-NPC, and a mention in the text of how large these

regions actually are.

The FADN is panel data – from which years were the data used? How many farms in each crop/region/type combination?

It should be stated in the data availability that the FADN data is partly confidential, and that the analyses can thus not be reproduced.

* On yield gap model:

(1) there is no consideration of alternative yield gap model specifications, or indeed a check that the assumptions of independence are correct. In fact, they are almost guaranteed to be false since the data includes repeated measures per farm. We are left in the dark as to how many farms lie behind the yield gap estimates.

(2) Including some form of interaction between techno-economic orientation and typology (organic/conventional) could be good because one could expect that yields are more similar in mixed farming, for instance, than in specialized crop farms. If organic farms only cover certain farm types I would suggest to restrict the input data to comparable farms rather than rely too much on the covariates to fix this. Given the large size of the FADN regions, I suspect that further grouping farms by techno-economic orientation (as an additional random effect) using that very same FADN data would help getting better estimates of the yield gaps which are central to this paper

(3) There is inclusion of mineral fertilizers in the model, but not of other types of fertilizers that are used in organic farming, i.e. manure, slurry, and the input of nitrogen through grass-clover leys. This could potentially cause a bias in the estimates of yield gaps. I expect that organic farmers may often be close to matching fertilization levels of their conventional counterparts.

(4) I would like to see model diagnostics (homoscedasticity, normality, leverage etc.). Also, it would be important to ensure that your explanatory variables are not too strongly correlated with each other so as to hamper the estimation of the effects (beta1). In fact, it is probably sufficient that the X1 variables are not too strongly correlated with the organic variable since that it is the coefficient for that variable that is used later on.

(5) There is cleaning of outliers going on in the code, that needs to be explained and motivated in the paper

(6) The final data only contains a subset of the regions

* on the NPC mixed model:

Model diagnostics are also needed here. The residuals are not independent of spatial location... There is a significant effect of the relative size of the ratio of evidence for organic to evidence for conventional on the estimate, as well as of the biogeographic variable. Together this reduces the estimate for the standardized LF_NPC by almost half, from 6.1 to 3.3. It is still significant (P=0.02). R code for that:

```
explore_lmer = lmer(estimate ~ I(nobs_organic/nobs)+biogeo+median_biogeo_norm + (1 | crop), data = npc_clean)
car::Anova(explore_lmer)
```

On CAPRI modelling:

There is insufficient detail to allow assessment of the way in which yield gaps estimates were introduced into the model. If I am not mistaken, the following is literally the only information available in the paper “In this study, we project all EU regions to reduce their pesticides use to levels observed under organic farming by 2030, and adopt the estimated yield gaps, taking into account each region’s NPC potential and the pesticides reduction. The latter is implemented by changing yields and pesticides cost under the regional agricultural supply modules of CAPRI”.

L370 which CAPRI supported stable release (STAR) was used?

Other comments

Abstract: The statement “6.1% increase in agricultural income” in the abstract (L14) is not correct, as this is the reduction in the yield gap observed (L134).

L31 replace "such as" with "including amongst others", as it is hard to have a complete list here (you missed bats and spiders, for instance)

L42-44 but see Karp et al. (cited in the study) which finds inconsistent effects.

L161 and L163 Provence-Alpes-Côte d'Azur

L183 it is not species richness but abundance (Rega et al. 2018 and papers cited therein) L309-311 how can sub-regional data be accessed if we do not know where the farms are within the FADN region?

L328 mention that the map is only considering agricultural areas.

Figure 4 - when "region" is used, mention which region it is (there are multiple ones in play in this study).

Yann Clough

Reviewer #3:

Remarks to the Author:

This study aims to quantify the benefits of landscape features that promote natural pest control (LF-NPC) for multiple crops across Europe. The authors estimate the yield gaps between organic production and conventional production and then use a map of LF-NPC potential in Europe to analyze the relationship between the yield gap and LF-NPC potential. They also use a model to predict the changes in yield gap associated with a 50% reduction in pesticides, which is the target reduction under the Biodiversity Framework for the year 2030. Finally, the study estimates how LF-NPC would affect agriculture income under that scenario. The results suggest that areas with high LF-NPC potential experience lower productivity losses due to pesticide reduction.

I find the approach presented in this paper very interesting, and the results are significant. The concepts are well-explained, and the paper is written very well. The major conclusions align with the results. However, I have a few points that I would like the authors to address to make this paper stronger.

1) The study does not account for the potential changes in biodiversity (and in turn on natural pest control) that could result from a decrease in pesticide use. Their model assumes that any fluctuations in biodiversity and natural pest control arise solely from landscape features, and not from the long-standing history of pesticide use in the area. As a result, the model appears to be static in this regard. It would be interesting to incorporate dynamic features that demonstrate how reducing pesticide usage could lead to an increase in biodiversity and contribute to NPC. It's possible that the impact of pesticide use on biodiversity is even greater than that of landscape features in highly intensive farming regions. Regardless of whether the authors choose to include this dynamic component in their model, they should discuss the potential impact of decreasing pesticide use on biodiversity.

2) Although the paper focuses on examining the impact of landscape-level features on biodiversity and NPC, it's crucial to consider the effect of local-level diversification as well. Intercropping, cover crops, rotations, silvopastoral systems, small-scale hedge rows, etc. are some of the examples of local-level diversification that can have a greater impact on increasing biodiversity and NPC than the landscape-level features. Therefore, the paper should also discuss the importance of local-level diversification and cite relevant literature examining its effects.

3) The model fails to consider an important factor which is the land that may need to be taken away from production to incorporate landscape-level features. This aspect is closely linked to the previous one where diversifying within plots through intercropping and other techniques can increase NPC without sacrificing the yield or losing land that is currently in use.

Minor comments:

- 1) Please provide clarification on the specific costs associated with reduced pesticide use as mentioned in lines 196-198. Additionally, please elaborate further on how the extent of the y-axis in relation to the data presented explains the point made in line 214. Thank you.
- 2) Figure 1 should include significance levels in addition to correlation coefficients.
- 3) Figure 3 should include the regression coefficient and significance value.

Replies to the comments of the reviewers

We would like to thank the three reviewers for their comments and suggestions, which have significantly helped to improve our manuscript. Point-by-point replies to the reviewer comments are inserted below in blue. Kindly note, while addressing your comments, we took the liberty of splitting the comments where needed, particularly when a single remark involved multiple amendments or covered several topics. This enabled us to tackle each comment with the appropriate attention and precision.

In addition to revising the text, we have made three major updates of the analysis in response to the received comments. These three most significant changes can be summarised as follows:

- 1) **LF-NPC regional score re-estimation:** We re-estimated the LF-NPC scores at regional level taking into account a proxy for where organic farms are located. In addition, we have also used a crop mask to re-calculate crop-specific LF-NPC scores, which provide a more granular regional LF-NPC assessment, aligned with the specific crop areas. The re-estimation of the regional crop-specific LF-NPC scores resulted in several major changes, including the re-assessment of the impact of LF-NPC on yield gaps. The re-estimation also implied a few minor adjustments in the text of the manuscript. For instance, when comparing the percentage changes in income between regions with similar LF-NPC profiles, we swapped the region of Bourgogne with Picardie, whose LF-NPC profile after the re-estimation became more similar to the region of Provence-Alpes-Cote d'Azur (previously used).
- 2) **Yield gap model update to account for additional management practices:** We updated our model specification to include additional covariates, capturing a broader spectrum of fertilizers, as well as varieties less sensitive to pests, and crop rotations. Furthermore, the revised approach involves running separate models per farm specialization. Consequently, we obtain yield gap estimates for each crop-region-specialization. We then compute the median of the estimated gaps across different specializations to get a crop-region specific yield gap. The new covariates and model structure are further explained in the Method section.
- 3) **Re-estimation of LF-NPC versus yield gaps:** We were able to identify two additional crops, for which we had sufficient data points to regress LF-NPC against yield gaps. This and the previous two amendments (LF-NPC re-estimation and yield gap model update) have led to a re-estimation of the mixed effect model associating a region's LF-NPC score to its yield gap. The modification resulted in a decrease in the coefficient associated to LF-NPC, from 6.1 percentage points to 4.4 percentage points per unit increase of LF-NPC. The refined results are now explained in more detail in the Results section.

Reviewer #1:

This is a well written and interesting paper mapping the magnitude of pest control across Europe and relating it to yield gaps if synthetic insecticides are avoided. Even though I find the approach commendable, I have several doubts about the pest control and yield gap estimates. To make these estimates more convincing **I would have liked to see validation with smaller datasets using real estimates of pest control (e.g. from sentinel prey cards and exclusion cages) and through real estimates of yield gaps due to pesticides when keeping everything else constant.**

>>> Reply

We appreciate your concerns regarding the validation of our pest control and yield gap estimates. We also agree that incorporating data from smaller, real-world experiments could strengthen our results. However, conducting such experiments would require significant time and resources, which were unfortunately beyond the scope of our current study. At the same time, some assumptions would be needed then to generalise results obtained in case studies with a narrower scope to perform an EU-wide assessment, as pursued in our paper. This generalisation could be especially difficult to perform at regional level as any region could claim its specificities. However, the aim of our paper is indeed obtaining a broader understanding of the relationship between LF-NPC and yield gaps across diverse regions and crop types. This aligns with the current gap in the literature, where a general LF-NPC quantification is missing.

Addressing your concerns about validation, we took several steps to fortify and validate our results concerning yield gap estimates attributed to pesticides. We reference existing literature reviews and conduct a comparative analysis with major meta-analyses that specifically focus on yield gaps between organic and conventional farming, including relevant field experiments. Additionally, we tried to identify relevant studies that look at yield gaps due to differences in pesticides use, a still less common topic in the literature, but nonetheless existing. Overall, we find that our yield gaps are in line with estimates provided in the literature (see L250 ff):

Furthermore, in response to your comment, we did an additional comparison with Bremmer et al. (2021), who conducted an expert survey, asking experts about the impact of pesticide reduction on yields for various crops. The following table presents the expert opinion on the yield reduction due to an assumed 50% reduction in overall use and risk of pesticides, along with a 50% reduction in the use of more hazardous pesticides, as estimated by Bremmer et al. (2021). In addition, we report our estimates of yield differentials resulting from pesticide reduction based on the organic-conventional gaps.

	Average yield change - Bremmer et al. (2021)	Average yield change - Own estimations
Wheat	-6	-23
Rapeseed*	-8	-23
Sugarbeet*	-12	-16
Maize	-2	-13

* Rapeseed and Sugarbeet are shown here, but excluded from the subsequent LF-NPC analysis due to limited data points. See Supplementary Information section two for more details on crop selection.

When comparing these estimates, crucial differences in the underlying assumptions need to be acknowledged. Bremmer et al. (2021) assume a reduction in line with Green Deal targets, whereas we estimate an almost complete reduction in pesticides use (a complete elimination in synthetic pesticides, which are not allowed under organic farming). Furthermore, the 50% reduction provides considerable flexibility regarding which active substances in pesticides to reduce (Schneider et al. 2023). Therefore, our estimates are significantly larger than those provided by Bremmer et al. (2021), with the exception of sugarbeet, for which even a 50% pesticides reduction is assumed to result in yields close to those of organic farming. Furthermore, in addition to the difference in the assumed pesticide reduction target, Bremmer et al. (2021) also needed to use additional assumptions to aggregate the farm-level evaluation results into EU Member State-level data for their EU-level simulation study. These assumptions were necessary to bridge the gap between micro-level and macro-level data. The Bremmer et al. (2021) example underlines that, even though a case study approach can provide important information, the ability to generalise results obtained with case studies may be limited by the specificities of the cases and the assumptions required for generalisation. Finally, regarding your suggestion of smaller datasets with real estimates of pest control, we acknowledge their importance, and future research could investigate in specific regions or crops using smaller-scale experiments or farm-level data, building upon our initial findings. As mentioned earlier, our study prioritizes obtaining a broader understanding of the overall relationship between LF-NPC and yield gaps beyond local field trials, aiming to identify general trends. This research effort is particularly driven by the observed lack of a general quantification of LF-NPC in the current literature on NPC.

Major comments:

1.1) I have doubts about the validity of the calculations of the study for three main reasons. First, the study estimates yield gaps attributable to synthetic pesticides by comparing yields in organic vs conventional fields. Purchase of mineral fertilizers and a number of other factors are accounted for and remaining differences in yields are assumed to be due to pesticide use. In my mind this still remains a fairly shaky assumption, since there are still lots of differences between farming systems not accounted for. Organic fields often use non-synthetic pesticides, may use varieties less sensitive to pests, use different crop rotations which may have strong effects on pests and enemies.

>>> Reply

We acknowledge your points of concern and have taken several steps to address them. To account for additional differences between organic and conventional systems we have included additional covariates in the yield gap models, which we now also referenced in the Method section L403 ff, along with the other covariates.

Manure

Specifically, in the revised version we use two variables related to fertilisation to account for non-synthetic fertilizers use: 1) fertilisers and soil improvers (in EUR) per hectare of utilised agricultural area (UAA), and 2) the total number of livestock units divided by the hectares of UAA.

The variable “fertilisers and soil improvers” represents the cost of all purchased fertilisers and soil improvers (e.g. lime) including compost, peat and manure (excluding manure produced on the

holding). This indicator divided by the total UAA is thus a proxy for the use of purchased fertilisers on the farm. Please note that this variable was already included in the previous estimation, however we did not correctly mention its definition. We corrected this in the revised manuscript and reemphasize here its use and definition. Moreover, it is important to emphasize that the variable fertiliser and soil improvers excludes manure produced on the holding. As there is no information regarding manure produced on the holding in the FADN survey, we assume a proportional relationship between manure use and the number of livestock units on the farm. Total livestock units are the sum of equines, cattle, sheep, goats, pigs, and poultry (annual average number), converted into a standardised measure, i.e., livestock units. Livestock units are determined by multiplying each animal category by a conversion coefficient representing the nutritional or feed requirement of each type of animal¹. The reference unit used for the calculation of livestock units (=1 LSU) is the grazing equivalent of one adult dairy cow producing 3,000 kg of milk annually, without additional concentrated feedstuffs. As feed requirements are correlated with manure production, the total number of livestock units per hectare proxies the amount of available manure on the holding that could be used as fertilizer. Additional information on FADN variables can be found at European Commission (2020a).

Varieties less sensitive to pests

To account for the use of varieties less sensitive to pests we include seed costs per hectare in the model. In the absence of qualitative information on seeds quality and their pest resistance, we assume that the use of varieties less sensitive to pest is correlated with the price of seeds. This assumption aligns with findings from the literature (Padel et al., 2021; Fernandez-Conejo and McBride, 2000; Lammerts van Bueren et al., 1999). In other words, we assume that varieties less sensitive to pests should be more expensive than other varieties. Consistent with the literature, the use of seed costs per hectare represents a reasonable strategy to account for the use of seeds of varying quality.

Non-synthetic pesticides

Regarding the use of non-synthetic pesticides, our approach aims at identifying differences in yields due to differences in crop protection practices between conventional and organic farms. Unfortunately, the FADN survey lacks specific information between synthetic and non-synthetic pesticide use, preventing a nuanced analysis in this regard.

Crop rotation

Furthermore, crop rotations are not straightforward to account for in the models. The reason is that the FADN sample of farm operates as a rotating panel and each farm may be present only in alternate years in the survey and not always in consecutive ones. Moreover, no field-level information is recorded in the survey and while information about the crop mix is available, it is not known where these crops have been grown on the farm. In addition, our sample of farms starts in 2010, and we do not have information on preceding cultivation years. In the absence of clear information on crop rotation, we proxy crop rotation with an index of crop diversity, using a Shannon index on shares of cultivated hectares by crop. Crop diversity is often used as a proxy of crop rotation in simulation studies using static positive mathematical programming based on FADN data (Kremmydas et al., 2024; Petsakos et al, 2022; Cortignani and Dono, 2020). To further support the use of crop diversity as proxy

¹ [https://ec.europa.eu/eurostat/statistics-explained/index.php?title=Glossary:Livestock_unit_\(LSU\)](https://ec.europa.eu/eurostat/statistics-explained/index.php?title=Glossary:Livestock_unit_(LSU)) accessed 3 January 2023.

for rotation, we investigated the relation between the Shannon index of crop diversity and organic status of the farms. We found that, on average, crop diversity is higher in organic farms than in conventional ones when farms specialize in field crops and horticulture. We run a linear model of the following form:

$$ShannonIndex_{it,k} = \alpha + \beta Organic_{it,k} + \gamma MemberState_{i,k} + \delta Year_{t,k} + \varepsilon_{it,k}$$

for $i = 1, \dots, N$; $t = 2010, \dots, 2017$; $k = 1, \dots, 8$.

where the Shannon Index for farm i at time t belonging to specialization k ($ShannonIndex_{it,k}$) is regressed against a set of explanatory variables, including a constant (α), the organic status of the farm ($Organic_{it,k}$), the member state of the farm ($MemberState_{i,k}$), the year of survey ($Year_{t,k}$) and an error term ($\varepsilon_{it,k}$). Standard errors of coefficients have been obtained by clustering errors at the farm level, thus allowing for both heteroscedasticity and autocorrelation of residuals. The resulting coefficients associated with each specialization are reported below:

Specialisation	Estimate	Std. Error	t value	Pr(> t)
Field crops	0.044	0.012	3.778	0.000
Horticulture	0.123	0.032	3.828	0.000
Wine	-0.023	0.021	-1.089	0.276
Other permanent crops	-0.110	0.013	-8.504	0.000
Milk	-0.261	0.014	-18.851	0.000
Other grazing livestock	-0.178	0.011	-16.018	0.000
Granivores	-0.045	0.036	-1.269	0.204
Mixed	-0.163	0.015	-11.002	0.000

These estimates show that for organic farms specialised in field crops and horticulture, the Shannon crop diversity index is on average significantly higher than for conventional farms. However, this is not the case for livestock-related farms and farms specialized in other permanent crops, for which the Shannon index of organic farms is on average smaller than that for conventional ones.

Stating limitations:

We acknowledge room for further improvement, which is now highlighted in the Discussion section of the manuscript (see L263 ff). Specifically, we recognize the potential of underestimating the yield gap due to data limitations in further accounting for management practices aimed at narrowing the productivity gap between organic and conventional farming. Similarly, we have included a section where we emphasize the importance of other local practices, which we do not capture in the yield calculation, neither through the LF-NPC indicator, which however may have an impact on the occurrence of pests and natural enemies.

L321 ff: "... local diversification strategies like intercropping, cover crops, and small-scale hedgerows (Jones et al., 2023), although context dependent, can offer farmers practical and timely solutions to pest management challenges (Rosa-Schleich et al., 2019). These practices are known for their potential to enhance the presence and effectiveness of beneficial insects and other natural enemies in agricultural ecosystems (Smith et al., 2020), without necessarily taken land away from agricultural

production. Therefore, such strategies can play a major role in reducing yield gaps in organic systems compared to conventional farming (Ponisio et al., 2015)."

1.2) Furthermore organic farms are often located in more extensive agricultural areas, and these differences may not be captured when averaging across regions.

>>> Reply

Your comment highlights a valid concern that we had not previously addressed due to the limited availability of data on the location of organic farms.

Following your comment, we were able to provide a solution by excluding areas from the regional LF-NPC estimation classified as intensive agriculture according to Rega et al. 2020. If areas categorized as low or medium intense are more likely to contain a higher percentage of organic farms, then the LF-NPC score of a region should be refined to that area. Implementing this refinement results in improved performance, confirming not only the validity of our initial findings, but suggesting as well the higher concentration of organic farms in low and medium intense areas. We have further elaborated on this approach and the performance in section one of the Supplementary Information.

Additionally, we would like to emphasize that in response to the suggestion from another reviewer, we have further refined the estimation of the regional LF-NPC utilizing a crop mask, enabling the generation of crop-specific LF-NPC values. We have provided a comprehensive explanation of this approach in the Supplementary Information section one and refer to it as well in the revised manuscript L70 ff.

Furthermore, within the yield gap estimation, we would like to emphasize that we have used a set of factors that are aimed at capturing locations of organic farms more specifically. These factors were already included in the models and no changes in this respect have been made in this revised version.

1.3) Second, the pest control estimates use flying invertebrate predators and parasitoids as a proxy. This approach has several limitations. It ignores ground dwelling predators which in some systems are key natural enemies and it ignores that pests respond very differently to landscape context (see e.g. Karp, D.S. et al. 2018. Crop pests and predators exhibit inconsistent responses to surrounding landscape composition. PNAS, 115 (33) E7863-E7870. <http://www.pnas.org/content/115/33/E7863>)

>>> Reply

Thank you for raising this important point. Indeed, using flying invertebrate predators and parasitoids as proxies for natural pest control has its limitations.

Ground-dwelling predators can significantly contribute to biocontrol in agroecosystems, helping to reduce both insect pests and weed seeds (Aviron et al., 2018; Mall et al., 2018). The abundance of ground-dwelling predators in arable crops has been shown to respond to the amount of grassland in the surrounding landscape (Aviron et al., 2018; Labruyere et al., 2016). However, research also suggests that soil management practices, which are not considered here, have a major influence on beneficial soil-dwelling species (Müller et al., 2022; Rowen et al., 2020). Conversely, actions related to

landscape structure, which is the focus of our paper, are known to provide resources to a broader range of beneficial species for suppressing pest populations (Chaplin-Kramer et al., 2011), although inconsistently (Karp et al., 2018).

The NPC map we used, originating from Rega et al. (2018), was built accounting for the abundance of natural enemies, as collected by the QUESSA project (Bartual et al., 2019; Rega et al., 2018). The QUESSA project is, to the best of our knowledge, the most comprehensive project at the pan-European scale in relation to this topic. The project QUESSA focused on a subset of case study countries (Italy, Switzerland, Germany, and the United Kingdom), containing a total of 217 SNHs sampled in 2013/2014, covering 38 herbaceous areal (HA), 61 herbaceous linear (HL), 55 woody areal (WA), and 63 woody linear (WL). The project primarily targeted 'flying predators,' which included:

- Dipteran families that either predate during their larval stage or actively hunt prey as adults. This group encompasses the families Asilidae, Dolichopodidae, and Syrphidae. It is worth mentioning that the subfamily Empidinae was excluded as they do not exhibit predatory behaviour.
- The guild of 'parasitic wasps' which are known for their parasitic lifestyle on a wide variety of organisms, including agricultural pests. This category includes hymenopteran superfamilies like Chalcidoidea, Braconidae, and Ichneumonidae.

Therefore, our study represents one approach to assess the potential impact of landscape features on pest control, using flying invertebrates as a simplified model, keeping in mind that landscape complexity increases not only flying invertebrates predators and parasitoids, but also a wide range of natural enemies. Consequently, while the index of Rega et al. (2018) may not be based on the complete spectrum of pest-natural enemy interactions, it provides a broad-scale estimation of natural enemy abundance and hence serve as a starting point for further research and discussion.

We have now included these considerations into the discussion, see L208 ff:

"Another critical aspect to consider is the utilization of flying invertebrates as a simplified model in this study to evaluate the potential of LF-NPC. The underlying data collection to construct the LF-NPC index focuses on flying invertebrates and not on the complete spectrum of natural enemies. For example, data on ground-dwelling organisms, known to significantly reduce insect pests and weed seeds in agroecosystems, were not collected (Aviron et al., 2018; Mall et al., 2018; Bartual et al., 2019; Rega et al., 2018). Hence, our study represents one approach to assess the potential impact of LF-NPC, using flying invertebrates as a simplified model. This approach is grounded in the understanding that landscape complexity enhances the diversity of natural enemies beyond flying predators and parasitoids. Therefore, the analysis provides a broad-scale estimation that can be used for further research."

Finally, with respect to the fact that pests respond very differently to landscape contexts. We reference these challenges, specifically the inconsistent results obtained in other studies in L47 ff. However, a strong benefit of our study is analyzing the relationship between yield gaps and landscape design across broader scales (across Europe) at regional resolution. This allows us to mitigate the limitations of more granular analysis which take into account for example local variability or specific pests-landscape complexities. Instead, we offer a general trend of the benefits and value of LF-NPC

for agricultural production, without undermining the importance of more granular analysis, as mentioned in L329 ff and L309 ff.

1.4) Third, the mapping is done across several different crops without adjusting to the relevant crop pests. Since pests respond differently to landscape composition and are accessible to different groups of natural enemies, flying natural enemies may not be a strong proxy for pest control across agricultural crops.

>>> Reply

We appreciate your valid point regarding the mapping of natural enemies and their potential impact on different crops.

To that purpose, we allow each crop group to have their individual relationship between yield gap and LF-NPC (see lines 128 ff on mixed effect model) by setting LF-NPC as a mixed effect per crop group. This approach enables crop-specific adjustments with respect to the impact of LF-NPC on yield gaps, which did not reveal any substantial differences. Therefore, we recognize the significance of delving deeper during the Discussion section with respect to the outcomes concerning the impact of insect pests within distinct crop groups. We have, therefore, included a dedicated section in our discussion:

L283 ff: "...the susceptibility of different crop groups to insect pests may vary. It is, however, challenging to provide a precise quantification of crop-specific damage caused by insects across different temporal and spatial contexts. This challenge arises from various factors, including the diverse agricultural environments in which crops are grown and the dynamic nature of insect pests, which can undergo cycles of emergence and re-emergence (Letourneau et al. 2017). An earlier study found that yield losses caused by insect pests range from 8 to 15% depending on the crop, with significant regional variations (Oerke et al. 2006). A more recent study analysing yield losses of major crops caused by pests and pathogens confirms the significant role of several insect pests in modern agriculture, similarly highlighting substantial regional variations (Savary et al. 2019). We accommodate for crop-specific susceptibility by allowing each crop to have an individual slope when relating yield gap to LF-NPC. However, this did not reveal substantial differences between crops. Additionally, interpreting the variation in correlation strength across crops between yield gaps and LF-NPC is challenging due to the lack of crop-specific quantifications of insect damage in organic farming. Despite this limitation, we can leverage the fundamental principle that insect pests have a natural predator or parasite and that organic farms in theory encounter the same insect pests as conventional ones but rely on natural processes for their control (Letourneau et al. 2017). Consequently, as insect pests remain key damage factors for crops (Savary et al. 2019), the positive correlation between yield gaps and LF-NPC aligns with the ecological rationale behind the potential of LF-NPC in mitigating severe pest infestations, thereby lowering the yield gap between organic and conventional farming."

Finally, as mentioned in the previous answer, our study focuses on using flying invertebrate predators and parasitoids as a simplified model to assess the potential impact of landscape features on pest control. Indeed, using flying natural enemies as a proxy for pest control across agricultural crops has its strengths and limitations. While it can provide valuable insights, it may not fully capture all aspects of pest control in diverse cropping systems. Pests can respond differently to landscape composition, and their *accessibility* to different groups of natural enemies can vary. Therefore, it is correct that our approach does not capture the full complexity of pest-natural enemy interactions across various agricultural crops and we have further acknowledged this limitation in the Discussion section L212 ff.

Nonetheless, and although we understand that in practice the effectiveness of NPC may vary depending on the crop and the specific pest, our approach assumes that if a landscape has a high potential for NPC, it inherently possesses as well the conditions to provide NPC for a broad range of pests. This assumption implies that such landscapes should naturally harbor a diverse range of natural enemies beyond those used as surrogate species, i.e. flying invertebrate predators and parasitoids, which should theoretically be independent of the specific crop or actual pest (see paragraph above in the reply to the previous comment).

2) Lines 124-129. These statements seem contradictory: 'The results indicate a statistically significant positive main effect of LF-NPC potential to yield gaps. Increasing a region's LF-NPC potential by one unit results in a 6.2% reduction in yield gap on average for all crops'. The first statement is supported by fig 2. The second statement sounds like the opposite pattern? According to the figures an increasing LF-NPC seems to result in an increase in the yield gap. Same interpretation on line 14-15. Is there something wrong here or am I missing something important?

>>> Reply

We acknowledge that the confusion might come from the fact that the yield gaps are expressed in negative numbers. To address this, we have added a clarifying statement in the manuscript:

*L133 ff: "The results indicate a statistically significant positive main effect of LF-NPC potential to yield gaps. **Despite the positive coefficient, it is important to note that yield gaps are expressed as negative numbers.** Therefore, an increase in a region's LF-NPC potential by one unit is associated with a 4.4 percentage points reduction in yield gap on average for all crops (Supplementary Table 4)."*

Minor comments:

3) Line 36-39. Not only seminatural habitats contribute to landscape complexity. Also crop diversity and field size can contribute strongly and both have been shown to be quite important for natural pest control.

>>> Reply

We agree with your comment. This aligns also with the comment made earlier on the importance of local conditions and other farm practices. Therefore, we further emphasize the importance of local conditions (see L305) and provide the two examples crop diversity and field size, which are well described in the cited literature.

4) Line 67. Do you mean 'flying insect predators and parasitoids'?

>>> Reply

Thank you for noticing this imprecision. We have corrected in L68: "... *flying insect predators and parasitoids.*"

5) Line 183, 185 and 189. Why 'species richness'?

>>> Reply

Yes, it is more correct to simply use species or natural enemies. We have corrected the text accordingly.

6) Figure SI 3. This is a table, not a figure.

>>> Reply

Thank you for noticing this imprecision. We have corrected accordingly.

7) Supplementary material line 58-59. This sentence is unclear. Something missing at the end?

>>> Reply

This has been corrected accordingly.

Additional references, mentioned in the replies to reviewer 1:

Aviron, S., Lalechère, E., Duflot, R., Parisey, N., & Poggi, S. (2018). Connectivity of cropped vs. semi-natural habitats mediates biodiversity: A case study of carabid beetles communities. *Agriculture, Ecosystems and Environment*, 268. <https://doi.org/10.1016/j.agee.2018.08.025>

Bartual, A. M., Sutter, L., Bocci, G., Moonen, A. C., Cresswell, J., Entling, M., Giffard, B., Jacot, K., Jeanneret, P., Holland, J., Pfister, S., Pintér, O., Veromann, E., Winkler, K., & Albrecht, M. (2019). The potential of different semi-natural habitats to sustain pollinators and natural enemies in European agricultural landscapes. *Agriculture, Ecosystems and Environment*, 279. <https://doi.org/10.1016/j.agee.2019.04.009>

Bremmer, J., Gonzalez-Martinez, A., Jongeneel, R., Huiting, H., Stokkers, R., Ruijs, M. (2021). Impact Assessment of EC 2030 Green Deal Targets for Sustainable Crop Production. Wageningen, Wageningen Economic Research, Report 2021-150. 70 pp.; 11 fig.; 33 tab.; 15 ref.

Chaplin-Kramer, R., O'Rourke, M. E., Blitzer, E. J., & Kremen, C. (2011). A meta-analysis of crop pest and natural enemy response to landscape complexity. In *Ecology Letters* (Vol. 14, Issue 9). <https://doi.org/10.1111/j.1461-0248.2011.01642.x>

Cortignani, R., and Dono, G. (2020). Greening and legume-supported crop rotations: An impact assessment of Italian arable farms. *Science of the Total Environment*, Volume 734, 10 September 2020, 139464, <https://doi.org/10.1016/j.scitotenv.2020.139464>

Dainese, M., Schneider, G., Krauss, J., & Steffan-Dewenter, I. (2017). Complementarity among natural enemies enhances pest suppression. *Scientific Reports*, 7(1). <https://doi.org/10.1038/s41598-017-08316-z>

EU (2013). REGULATION (EU) No 1305/2013 OF THE EUROPEAN PARLIAMENT AND OF THE COUNCIL of 17 december 2013 on support for rural development by the European Agricultural Fund for Rural Development (EAFRD) and repealing Council Regulation (EC) No 1698/2005.

European Commission (2020a). Farm Return Definitions. RI/CC 1680 v6, Brussels, May 2020.

European Commission (2020b). Farm Accounting Data Network. An A to Z Methodology. Version 31/10/2020. At <https://circabc.europa.eu/ui/group/befb6055-ab0c-4305-84fe-0c80c1c0553d/library/1df3a121-11ee-40c3-a991-70a5f3cdd9d7/details>.

Fernandez-Conejo, J. and McBride, W.D. (2000). Genetically Engineered Crops for Pest Management in US.S Agriculture: Farm-Level Effects. Resource Economics Division, Economic research Service U.S. Department of Agriculture, Agricultural Economic report No. 786.

Karp, D. S., Chaplin-Kramer, R., Meehan, T. D., Martin, E. A., DeClerck, F., Grab, H., Gratton, C., Hunt, L., Larsen, A. E., Martínez-Salinas, A., O'Rourke, M. E., Rusch, A., Poveda, K., Jonsson, M., Rosenheim, J. A., Schellhorn, N. A., Tscharrntke, T., Wratten, S. D., Zhang, W., ... Zou, Y. (2018). Crop pests and predators exhibit inconsistent responses to surrounding landscape composition. *Proceedings of the National Academy of Sciences of the United States of America*, 115(33). <https://doi.org/10.1073/pnas.1800042115>

- Kremmydas, D., Ciaian, P., Baldoni, E. (2024). Modeling conversion to organic agriculture with an EU-wide farm model. *Bio-based and applied economics*, 10.36253/bae-13925.
- Labruyere, S., Bohan, D. A., Biju-Duval, L., Ricci, B., & Petit, S. (2016). Local, neighbor and landscape effects on the abundance of weed seed-eating carabids in arable fields: A nationwide analysis. *Basic and Applied Ecology*, 17(3). <https://doi.org/10.1016/j.baae.2015.10.008>
- Lammerts Van Bueren, E.T., Hulscher, M., Haring, M., Jongerden, J., van Mansvelt, J.D., den Nijs, A.P.M., Ruivenkamp, G.T.P. (1999). Sustainable organic plant breeding. Final report: a vision, choices, consequences and steps. Louis Bolk Instituut, natuurwetenschappelijk onderzoek.
- Letourneau, D., & Bruggen, A. van. (2017). Crop protection in organic agriculture. In *Organic agriculture: a global perspective*. <https://doi.org/10.1079/9781845931698.0093>
- Litterick, A. M., Watson, C. A., & Atkinson, D. (2002). Crop protection in organic agriculture – a simple matter? Proceedings of the UK Organic Research 2002 Conference: Research in Context. 26-28th March 2002 Aberystwyth.
- Mall, D., Larsen, A. E., & Martin, E. A. (2018). Investigating the (Mis)match between natural pest control knowledge and the intensity of pesticide use. *Insects*, 9(1). <https://doi.org/10.3390/insects9010002>
- Müller, P., Neuhoff, D., Nabel, M., Schiffers, K., & Döring, T. F. (2022). Tillage effects on ground beetles in temperate climates: a review. In *Agronomy for Sustainable Development* (Vol. 42, Issue 4). <https://doi.org/10.1007/s13593-022-00803-6>
- Oerke, E. C. (2006). Crop losses to pests. In *Journal of Agricultural Science* (Vol. 144, Issue 1). <https://doi.org/10.1017/S0021859605005708>
- Padel, S., Orsini, S., Solfanelli, F., Zanoli, R. (2021). Can the Market Deliver 100% Organic Seed and Varieties in Europe? *Sustainability* 2021, 13, 10305. <https://doi.org/10.3390/su131810305>.
- Petsakos, A., Ciaian, P., Espinosa, M., Perni, A., and Kremmydas, D. (2023). Farm-level impacts of the CAP post-2020 reform: A scenario-based analysis. *Applied Economic Perspectives and Policy*, Volume 45, Issue 2, <https://doi.org/10.1002/aepp.13257>.
- Rega C., Short C., Pérez-Soba M., Luisa Paracchini M. (2020). A classification of European agricultural land using an energy-based intensity indicator and detailed crop description. *Landsc Urban Plan.* 2020;198. doi:10.1016/j.landurbplan.2020.103793
- Rowen, E. K., Regan, K. H., Barbercheck, M. E., & Tooker, J. F. (2020). Is tillage beneficial or detrimental for insect and slug management? A meta-analysis. *Agriculture, Ecosystems and Environment*, 294. <https://doi.org/10.1016/j.agee.2020.106849>
- Savary, S., Willocquet, L., Pethybridge, S. J., Esker, P., McRoberts, N., & Nelson, A. (2019). The global burden of pathogens and pests on major food crops. *Nature Ecology and Evolution*, 3(3). <https://doi.org/10.1038/s41559-018-0793-y>
- Schneider, K., Barreiro-Hurle, J., Rodriguez-Cerezo, E. (2023). Pesticide reduction amidst food and feed security concerns in Europe. *Nature Food* 4, 746–750. <https://doi.org/10.1038/s43016-023-00834-6>

Reviewer #2 (Remarks to the Author):

This study attempts to estimate crop yield and economic outcomes of reducing pesticide use in Europe, taking into account the fact that the natural pest control is likely to depend on the landscape features. An ambitious and timely approach given the environmental commitments at international, EU and often also national levels. The "trick" the authors are using is assuming that organic yields (controlling for synthetic fertilizer use) represent the yield levels under conventional agriculture with reduced pesticide use. Yield gaps [organic-conventional] are calculated and regressed against an index for landscape-features supporting natural pest control (LF-NPC) at the level of the FADN regions (FADN is a state-of-the art farm panel dataset), controlling for other farm-level characteristics. The results suggest that the yield gap is reduced as LF-NPC increases. This is as far as I know a novel and very interesting result in itself, even though I lack a discussion of why LF-NPC should be a good predictor for yield (gaps), and a more critical approach when it comes to the modelling (see comments below).

In a second step, to simulate what would happen under a drastic reduction of pesticide use the authors introduce the yield gaps and reductions in pesticide costs into the CAPRI model, a state-of-the art European agricultural sector model. They find that the income changes are also reduced – incidentally by almost exactly the same amount - if there is a higher LF-NPC in the (FADN) region, partly through the yield gap effect, and partly through the difference in pesticide use between regions and crops.

A very interesting study overall, yet I have a number of concerns that I think need to be addressed before this study can be reassessed.

* on the level of detail in terms of methods and results

I would like to see much more detail on results, model diagnostics, removal of outliers, which regions were or were not included in the end. This is important to assess the validity of the approach. See comments below for more specifics.

* On using organic as pseudo-treatment for conventional with no (or much less) pesticides

Organic farming often is different from conventional farming in many ways. While covariates have been used for the estimation of yield gaps to reduce that problem, issues remain. For example, organic farmers are often maintaining soil fertility and suppressing weeds by having longer rotations (that often include grass-clover leys) than conventional farmers – unclear if this is taken into account in the CAPRI part of the study, I think it is just the yields that are changed not the rotations. I would like to see a version of the CAPRI modelling where rotations are also adapted. I think this could give a range of what we would expect in future if pesticide use is reduced significantly.

>>> Reply

We appreciate your feedback and suggestion, which partially complements the insights offered by another reviewer. Before anything else, we agree with your observation that yield gaps can be influenced by a very large range of farm practices and that reducing pesticide use could potentially lead to a shift in farm management practices, such as increased crop rotation.

As regards CAPRI, the model considers specific rotation requirements as outlined in the Common Agricultural Policy. While such rotations may be affected in the scenario runs, we do not consider a specific longer rotation pattern that would be related to an increase in organic farming. However, we think that this is not necessary, because it is not the aim of the analysis in this paper to mimic an increase in rotation pattern. What we are aiming at is to obtain a broader understanding of the relationship between LF-NPC and yield gaps across diverse regions and crop types. We, therefore, also solely want to investigate LF-NPC and its associated yield impact during the CAPRI simulation. We apply CAPRI to assess the potential economic value associated with LF-NPC in terms of its contribution to agricultural income in an environment without synthetic pesticides use. Accordingly, in the simulation with CAPRI we only use the yield shocks (i.e. yield benefits) attributable to LF-NPC (as estimated in the major part of the paper).

Although it is only indirectly linked to your comment, we also expanded on the topic of crop rotation with respect to the yield gap estimation, a suggestion raised by reviewer 1. Here, a replication of our reply to reviewer 1, where we acknowledge that the impact of crop rotations and other farm management practices should be ideally further incorporated into the yield gap calculation, to truly isolate their effect and capture the yield gap attributable to differences in pesticides use.

While FADN data constraints pose a challenge for controlling all farm practices in the yield gap estimation, the issue of crop rotation becomes particularly evident as the same farms are not reported across years. The reason is that the FADN sample of farm is a rotating panel, and each farm may be present only in alternate years in the survey and not always in consecutive ones. Moreover, no field-level information is recorded in the survey and while information about the crop mix is available, it is not known where these crops have been grown on the farm. On top of that, our sample of farms starts in 2010 and we do not have information regarding preceding years of cultivation. In the absence of clear information on crop rotation, we proxy crop rotation with an index of crop diversity. We use a Shannon index on shares of cultivated hectares by crop. Crop diversity is often used as a proxy of crop rotation in simulation studies based on static positive mathematical programming based on FADN data (Kremmydas et al., 2024; Petsakos et al, 2022; Cortignani and Dono, 2020). To further support the use of crop diversity as proxy for rotation, we investigated the relation between the selected proxy for crop diversity and organic status of the farms. We found that, on average, the chosen indicator for crop diversity is higher in organic farms than in conventional ones when farms are specialize in field crops and horticulture. We run a linear model of the following form:

$$ShannonIndex_{it,k} = \alpha + \beta Organic_{it,k} + \gamma MemberState_{i,k} + \delta Year_{t,k} + \varepsilon_{it,k}$$

for $i = 1, \dots, N$; $t = 2010, \dots, 2017$; $k = 1, \dots, 8$.

where the Shannon Index for farm i at time t belonging to specialization k ($ShannonIndex_{it,k}$) is regressed against a set of explanatory variables. These include a constant (α), the organic status of the farm ($Organic_{it,k}$), the member state of the farm ($MemberState_{i,k}$), the year of survey ($Year_{t,k}$) and an error term ($\varepsilon_{it,k}$). Standard errors of coefficients have been obtained by clustering errors at the farm level thus allowing for both heteroscedasticity and autocorrelation of residuals. The resulting table of coefficients β associated to each specialization k is reported here:

Specialisation	Estimate	Std. Error	t value	Pr(> t)
----------------	----------	------------	---------	----------

Fieldcrops	0.044	0.012	3.778	0.000
Horticulture	0.123	0.032	3.828	0.000
Wine	-0.023	0.021	-1.089	0.276
Other permanent crops	-0.110	0.013	-8.504	0.000
Milk	-0.261	0.014	-18.851	0.000
Other grazing livestock	-0.178	0.011	-16.018	0.000
Granivores	-0.045	0.036	-1.269	0.204
Mixed	-0.163	0.015	-11.002	0.000

These estimates show that for organic farms specialised in field crops and horticulture, the Shannon crop diversity index is on average significantly higher than for conventional farms. This is not the case for livestock related farms and for farms specialized in other permanent crops for which the Shannon index of organic farms is on average smaller than that of conventional ones.

In light of these results and consistent with reviewer 1's comments, we acknowledge the importance of explicitly acknowledging the potential for further refinement in the yield gap estimation with respect to farm management practices. In the discussion, we have added:

L264 ff: "While the estimated yield gaps observed in this study are consistent with those reported in previous research, it is important to note that the model structure and covariates used may lead to an underestimation of the true yield gap between organic and conventional due to differences in pesticides practices. Specifically, if management practices that can help to bridge the productivity gap between organic and conventional farming are not fully captured, they might nuance the yield gap estimation and result in lower than actual yield gaps. In this study those management practices in some cases could only be approximated, such as the selection of high-yielding organic seed varieties or the implementation of effective crop rotation practices (see Method section)."

* On the LF-NPC data and its use and communication:

There is little discussion about the ecological motivation for using LF-NPC for explaining yield gaps, even though Rega et al. 2018 are very careful about discussing limitations in interpretation and onward use. Is there other evidence suggesting that yield gaps between conventional and organic are due to pests/pesticide use?

>>> Reply

We recognize the importance of further explaining the ecological motivation of employing LF-NPC to explain yield gaps and the connections between yield gaps and pesticide usage. Therefore, we have incorporated the following paragraph into the Discussion section:

L276 ff: "With organic agriculture defined as a farming system that does not use synthetic pesticides or mineral fertilizers (Seufert et al. 2017), reduced yield performance in organic farms can be attributed to nutrient limitations, pests and diseases. Pests and diseases play a more significant role in organic farming than in conventional farming, given the limited options for rapid nutrient replenishment or pest control through pesticides application (De Ponti et al. 2012, Knapp et al. 2018). Consequently, pest management in organic farming relies on natural processes (Seufert et al. 2012), such as the occurrence of natural enemies, whose abundance can be influenced by both management practices and landscape complexity (Bianchi et al. 2006)."

Moreover, the discussion is further enriched by discussing the crop-specific component of pest susceptibility, which follows the suggestion of Reviewer 1:

L283 ff: " ... the susceptibility of different crop groups to insect pests may vary. It is, however, challenging to provide a precise quantification of crop-specific damage caused by insects across different temporal and spatial contexts. This challenge arises from various factors, including the diverse agricultural environments in which crops are grown and the dynamic nature of insect pests, which can undergo cycles of emergence and re-emergence (Letourneau et al. 2017). An earlier study found that yield losses caused by insect pests range from 8 to 15% depending on the crop, with significant regional variations (Oerke et al. 2006). A more recent study analysing yield losses of major crops caused by pests and pathogens confirms the significant role of several insect pests in modern agriculture, similarly highlighting substantial regional variations (Savary et al. 2019). We accommodate for crop-specific susceptibility by allowing each crop to have an individual slope when relating yield gap to LF-NPC. However, this did not reveal substantial differences between crops. Additionally, interpreting the variation in correlation strength across crops between yield gaps and LF-NPC is challenging due to the lack of crop-specific quantifications of insect damage in organic farming. Despite this limitation, we can leverage the fundamental principle that insect pests have a natural predator or parasite and that organic farms in theory encounter the same insect pests as conventional ones but rely on natural processes for their control (Letourneau et al. 2017). Consequently, as insect pests remain key damage factors for crops (Savary et al. 2019), the positive correlation between yield gaps and LF-NPC aligns with the ecological rationale behind the potential of LF-NPC in mitigating severe pest infestations, thereby lowering the yield gap between organic and conventional farming."

Weeds and fungal diseases may be a more important limiting factor and are not really covered by LF-NPC I think.

>>> Reply

The main factors leading to yield gaps between organic and conventional farming, primarily imposed by legislative restrictions, are from restrictions on chemical fertilisers and pesticides use (De Ponti et al. 2012 and European Parliament and Council of the European Union 2018). These restrictions, in turn, can lead to differences in other management practices. We tried to isolate as much as possible the yield gap effect attributed to the differences in pesticide use by controlling specifically for fertilisers in the yield gap estimation, and accounting for management practices and pedoclimatic conditions (as detailed above and in the Method section).

Concerning the importance of weeds and fungal diseases: Organic agriculture employs relatively effective methods for controlling weeds, as well as fungal and bacterial diseases. Fungal and bacterial diseases can be controlled with substances such as copper, sulphur, aluminium sulphate, potassium hydrogen carbonate and lime sulphur. For example, copper is currently approved as an active substance in plant protection products (PPP) for more than 50 different bacterial and fungal diseases in viticulture, horticulture, hops, market garden and arable crops in the EU (IFOAM 2018 and Tamm et al. 2022). The European Commission allows up to 6kg/ha and year of copper, and in many European countries copper is used up to this limit (CO-FREE Consortium 2016). As for weeds, their control in organic farming also counts on several non-chemical strategies, such as cropping system design and rotation, mechanical weed management techniques, soil management practices, breeding, bio-based herbicides, precision and site-specific weeding and digitalised weed monitoring, including using technology and knowledge of weed biology (Fogliatto et al. 2023). For controlling insects in organic

farming, the amount of available and allowed techniques (e.g. biological control) and substances are limited, generally species-specific, not being effective for other insects. Even though insects can be considered one of the most important limiting factors in crop protection leading to yield gaps between conventional and organic farming, we acknowledge that only part of the gap attributed to non-use of pesticides from our results is caused by insects. Thus, we could expect that the explanatory potential of LF-NPC in explaining yield gaps could be further refined by incorporating additional ecosystem services specifically targeting weeds and fungal diseases.

Furthermore, it is important to note that the FADN database does not separate the crop protection variable into different sub-categories, which prevents a more precise analysis. If sub-categories were available, and we could access information about the gap attributed to insecticides only, we would expect an increased significance of the effect of LF-NPC on yield gaps.

I would recommend computing a crop-specific LF-NPC estimate for each region using the freely available EU crop type map for 2018. Indeed some of the crops may be grown in parts of the FADN region only, thus biasing the LF-NPC estimates for a particular crop away from their true value?

>>> Reply

We followed the reviewer's recommendation, considering it an excellent idea to use LF-NPC scores which are further fine-tuned to where the specific crop is actually grown. Initially, the LF-NPC map was tailored to agricultural areas. Now, we focus on the exclusion of LF-NPC pixels where the targeted crop is absent. This refinement involved employing the EU crop type mask at 10-meter resolution you refer to with one caveat: Since we are using yield gap observations between 2010 to 2017 (see L98), it is more appropriate to upscale the EU crop mask to 10 kilometer (km) and assess whether a specific crop is present. Each 10x10 km pixel was assigned a value of 1 if the crop was present and 0 otherwise. This resultant 10x10 km crop mask, representing the crop's presence or absence, was then applied to the LF-NPC map to filter out pixels where the crop was not present. By upscaling to 10 km resolution, we aim to generalize the 2018 map to earlier years covered in the study but lacking a detailed EU crop mask. Thus, we assume that the number of pixel classified as a certain crop in 2018 within a 10 km radius is the same as in previous years. The 10 x 10 km pixels lacking the crop of interest were excluded from the regional LF-NPC score calculations, thus enabling the estimation of crop-specific LF-NPC scores.

We explain this approach with further details in section one of the Supplementary Information. Additionally, we provide an Excel spreadsheet illustrating LF-NPC scores considering agricultural land (from the previous version), areas of low and medium intensity (as per the suggestion of another reviewer), and crop-specific areas. Notably, the correlation between yield gaps and LF-NPC was strongest when employing the crop-specific LF-NPC focusing on low and medium intense areas.

The heavy referring to Rega et al. 2018 makes understanding and communication of the results more complicated than necessary. In the Abstract, the statement "a one-unit of LF-NPC potential, on average, leads to a 6.1% increase in agricultural income" is meaningless if the reader does not get an idea of what that unit represents. I guess this can be different things, but in that case replacing by an example would be better. If length is a concern, cut down a bit on the linear mixed models details, with which many readers will be familiar).

>>> Reply

Thank you for the comment. Indeed, we understand the importance of providing additional details to clarify the meaning of a "unit" increase, considering that the index capturing LF-NPC potential might not be familiar to readers, especially at the beginning when reading the abstract.

To address this concern, we have rephrased the abstract to provide a more accessible interpretation of the results, avoiding the reference to the one-unit increase. This adjustment aims to offer meaningful insights to the reader from the outset. Furthermore, we have expanded on the interpretation of our findings in the results section. Specifically, we provide more information on the LF-NPC score range across Europe and relate the coefficient obtained from the mixed model (expressed as "one-unit increase of LF-NPC") to the average and maximum observed scores of LF-NPC across Europe. In the results section we now explain:

*L137 ff: "To comprehensively interpret the significance of these findings, it is crucial to look at the variability and range of the LF-NPC scores across Europe. Generating crop-specific LF-NPC scores and normalizing by biogeographic zones, LF-NPC scores range from 0 to a maximum of 2.9 across Europe. The majority of regions cluster within the scores of 0 to 1.5, while the European average settles at 0.9. Notably, the European average suggests that between 2010 and 2017, LF-NPC accounted for a 4 percentage point (0.9*4.4%) reduction in yield gaps between organic and conventional farming practices, on average. Furthermore, our results suggest that regions characterized by the maximum LF-NPC experiences a yield gap reduction of 12 percentage point (2.9 * 4.4%)."*

Further addressing your point, and making use of the more elaborated interpretation, we have revised the abstract to eliminate the reference to a one-unit increase of LF-NPC potential. Instead, we now emphasize the average and maximum impacts of LF-NPC, enhancing the accessibility of the interpretation for the reader. The modified abstract now reads:

L14: "Our analysis suggests that LF-NPC reduces yield gaps on average by four percentage points, and increases income by a similar magnitude."

Finally, we found it would also be beneficial to include detailed information in the Supplementary Information on the LF-NPC scores themselves (see section 1 of the Supplementary Information). The first part of the LF-NPC section offers a more comprehensive explanation on the original LF-NPC indicator, the underlying concept, a graph depicting its distribution, a justification for normalizing by biogeographic region, and an example of landscape characteristics for three regions with distinct LF-NPC scores. In the second part, we delve into the process of obtaining LF-NPC scores while excluding intensive agricultural areas and then proceed to explaining the process of obtaining crop-specific LF-NPC scores using a crop-mask.

We believe that these changes simplify and improve the communication of our results, enabling a more comprehensive grasp of the LF-NPC indicator and its ramifications.

This is compounded by the fact that it is not actually LF-NPC that is used but the LF-NPC standardized by biogeographic region. This is not well detailed anywhere, not even in the code. L328 L330 the

standardization of the LF-NPC needs to be explained more in detail. Why is it done and does the use of unstandardized data give very different results?

>>> Reply

We have provided further details in the supplementary material on the rationale behind normalizing by biogeographic region.

L26 ff SI: *“Biogeographic regions are categorized based on their biotas (Morrone 2018), encompassing the collection of all living organisms, including insects and microorganisms. Insects within a certain biota characterizing a biogeographic region may have different landscape requirements. Consequently, comparing the regional score of LF-NPC from different biogeographic regions could be misleading. To facilitate meaningful comparisons, a region’s LF-NPC score should be normalized by the median LF-NPC score of its corresponding biogeographic region. This normalization process ensures that the LF-NPC values are comparable across regions, irrespective of their biogeographic context.”*

Regarding the comment on whether the results change: We do see an improvement in the performance when normalizing by biogeographic regions. Considering the explanation provided above on this normalization choice, we believe it is important to present the results after the LF-NPC scores have been normalized by geographic region.

* On the FADN data:

Motivation for the use of FADN regions: I would like to see some evidence for the variability within and between FADN regions of the yields and LF-NPC, and a mention in the text of how large these regions actually are.

>>> Reply

FADN regions have been chosen because of the sampling design of the FADN survey. FADN regions are composed of Eurostat NUTS3 regions and represent one of the strata upon which the FADN survey was designed. The other two strata are economic size of farms and type of farming (TF) (European Commission, 2020b). This being said, it is important to note that FADN regions are not the smallest level of aggregation under the FADN database; NUTS3 is the most granular level. In fact, we use NUTS3 dummies in the yield gap estimation (see L390). Although it is possible to aggregate data at the NUTS3 level, it is not advisable due to the potential loss of statistical quality. Aggregating at this level may result in the exclusion of many NUTS3 regions due to insufficient data, and similarly many of the remaining models would be made of very few data points for organic farms, which is why it is not recommendable to aggregate at NUTS3 level.

Regarding the variability within regions for LF-NPC and yield gaps, we provide the requested details in an Excel spreadsheet given the large amount of regions (see 'variation_within_FADN_region.xlsx'). This spreadsheet shows the variability of LF-NPC and yield gaps for different crops across various tabs in the Excel file.

Regarding the variability across regions, we now provide distribution charts for both variables LF-NPC and yield gaps. Please refer to Figures SI1 and SI2 for a detailed overview of LF-NPC scores across regions, which have been integrated into the comprehensive discussion of the LF-NPC indicator.

Concerning the distribution of yield gaps across regions and crops, we have enhanced Figure 1 in the main text by employing histograms instead of the previous distribution charts, providing a more visually effective representation of the data.

Finally, regarding the question on how large these FADN regions are we have added this information in the Supplementary Information:

L18 ff SI: "A total of 133 FADN regions exist across the European continent, with the smallest region located in Belgium at 133 square kilometers to the largest region in Sweden at 261,956 square kilometers. The distribution of these regions follows a skewed pattern, with the 75th percentile falling at around 38,900 square kilometers."

Kindly note, that the total of FADN Regions is actually 139, however 6 shapes are located outside the European continent, such as for example Guadeloupe, Martinique or La Reunion.

The FADN is panel data – from which years were the data used? How many farms in each crop/region/type combination?

>>> Reply

We use the FADN data from 2010 to 2017. We have further specified in the text:

L98 ff: "Using data from 2010 to 2017, yield gaps are estimated for each FADN region of the EU and by farm typology, provided that a minimum number of organic farms per farm typology and crop operate in a region."

Additionally, the Excel file 'input_npc_yield_gap_data.xlsx,' documents under columns F the number of total farms used and under columns G the number of organic farms used for each yield gap estimation (crop/region/type - specific).

It should be stated in the data availability that the FADN data is partly confidential, and that the analyses can thus not be reproduced.

>>> Reply

We have added:

L510 ff: FADN - EU Farm Accountancy Data Network: https://agriculture.ec.europa.eu/data-and-analysis/farm-structures-and-economics/fadn_en. Kindly note that this data is confidential: *"According to Regulation (EC) No 1217/2009 the data are covered by strict confidentiality rules and can only be used to meet the needs of the common agricultural policy. For example, they cannot be used by authorities for tax or compliance purposes."*

* On yield gap model:

(1) There is no consideration of alternative yield gap model specifications, or indeed a check that the assumptions of independence are correct. In fact, they are almost guaranteed to be false since the data includes repeated measures per farm. & (4) I would like to see model diagnostics (homoscedasticity, normality, leverage etc.).

>>> Reply

In the Supplementary Information, we now explain in more detail the modelling steps, the assumptions taken, and diagnostics employed. In this revised version, we have refined the estimation approach and intersected techno-economic orientation with organic status of farms (as explained above, following another suggestion). Moreover, we carry out diagnostic tests for our models to check for homoscedasticity, normality, serial correlation, and potential influential observations.

In terms of diagnostics, first we check for homoscedasticity, normality and serial correlation. However, it is important to note that the three assumptions (homoscedasticity, normality, no autocorrelation) are relevant for the inference on the coefficient estimates of the yield gaps, but not for the estimates themselves. As it will be explained below, coefficient estimates are not affected (in our case) by these assumptions, as OLS would still be the preferred estimator, even when these assumptions fail. In other words, yield gaps in this study are not selected based on a threshold that defines statistical significance. Instead, they are taken as averages and used in subsequent modelling steps. We argue that this approach does not influence significantly the analysis while allowing an extended geographical scope that is important, given the limited number of sampled organic farms.

Using robust estimation (see below for details on robust estimation), we find that 60% of the estimated p-values of the organic dummy are significant at a 95% confidence level, with 66% of the estimated p-values showing significance at a 90% confidence level. Notably, adjusting the statistical significance threshold to lower confidence levels leads in a substantially higher number of yield gap estimations. The graph below shows the relationship between confidence levels and share of statistically significant estimates. This nuanced exploration of significance levels contributes to a more comprehensive understanding of the robustness of our findings, acknowledging variations in confidence thresholds.

By increasing our tolerance and including estimates with an associated p-value equal to 0.15, we would increase the number of statistically significant coefficients from 66% to 70%. By increasing it to

0.2, we would get around 74% statistically significant estimates, and setting it at 0.3 would get approximately 80% significant estimates. The graph shows that the relationship between confidence and share of significant observations strongly increases. As a consequence, filtering out yield gaps based on a pre-defined threshold of p-value (for example, larger than 0.1) appears arbitrary, especially when adjusting it to 0.15 significantly increases the number of available yield gap coefficients, enabling a broader geographical analysis. This is particularly relevant given the limited number of organic farms per crop, region and farm specialization. Low statistical significance may be a consequence of small sample size, as shown in the following distribution of organic farm used per estimated model, where models with less than 16 organic farms are excluded from the analysis.

For these reasons, in the first step of this study, when estimating the yield gaps, we do not filter coefficients based on their p-value. Nevertheless, to provide evidence of the (non)-effect of our choice, we did perform a sensitivity analysis, revealing that the consequences of our approach are limited in magnitude but beneficial for extending the analysis to larger geographical areas. In the accompanying scatterplots, we exclusively employ yield gap estimations with a significance level of 90%. Notably, the overall trend remains consistent across all crops, except for corn where the correlation between LF-NPC and yield gap disappears when focusing on the selected yield gaps. For the remaining crops, however, the positive overall correlation persists.

Despite not explicitly considering the statistical significance of coefficients, we provide evidence on the model diagnostics below.

Model diagnostics:

We use the Breusch-Pagan test for heteroscedasticity, the Breusch-Godfrey test for serial correlation, and the Shapiro test for normality of residuals after OLS estimation.

- Heteroscedasticity: 40% of the estimated models have a p-value of the Breusch-Pagan test larger than 0.1 (residuals are homoscedastic), increasing to 48% by setting the 0.05 p-value threshold.
- Serial correlation: 66% of the estimated model have a p-value of the serial correlation test higher than 0.1, reaching 74% considering a p-value of 0.05.
- Normality: around 17% of the estimated models shows a higher than 0.1 p-value (normality) for the Shapiro test, decreasing to 10% at a 0.05 p-value threshold.

Overall, these tests show that the assumption of normality of residuals is often not satisfied, homoscedasticity is satisfied in a share of models ranging between 40 and 48%, while most models do not show serial correlation in their residuals.

As additional diagnostics on residuals, we explore graphically their distribution after pooling all the regional errors together and displaying them by crop. As testing for normality in large samples is especially complex and unreliable (Royston, 1982, 1995), we test it visually using histograms and a normality plot, or QQ plot (Wilk and Gnanadesikan, 1986):

Besides the case of fodder corn, the histograms present a certain degree of symmetry.

With respect to the Q-Q plots, if the error distributions were normally distributed, points would align along the bisecting line of the plot. In the case of our distributions, the comparison of sample quantiles (y-axis) with theoretical quantiles (x-axis) shows the presence of some skewness and heavier-than-normal tails. In the case of fodder corn, legumes and peas, distributions present some skewness to the right while the rest of the distribution is close to a normal distribution.

If the hypothesis of homoscedasticity and autocorrelation were satisfied, OLS would be both consistent and most efficient (BLUE: best linear unbiased estimator) among the linear estimators. If these hypotheses were not tenable, one should perform GLS rather than OLS, which would be more efficient (BLUE) in this case. However, due to the lack of a known structure for the variance-covariance matrix of the residual distribution, the empirical feasibility of GLS is limited. This structure has to be estimated, and the only feasible option to implement GLS is to adopt a two-step approach where first,

an estimate of the variance-covariance matrix of residuals is obtained and then, in a second step, this estimate is used to perform FGLS (feasible GLS). For the estimation of the variance-covariance matrix of the residual distribution of the first step, OLS should be used as it is a consistent and implementable estimator of the coefficients. Despite GLS being more efficient than OLS, in small to medium-sized samples, as in our cases, FGLS may be less efficient. Moreover, there may be cases in which FGLS is not consistent and this poses a risk for the estimation of the yield gaps. In fact, very little is known about the small sample properties of FGLS (Hayashi, 2011). Given these trade-offs in implementing FGLS, standard practice, when standard hypotheses fail, is to use a consistent estimator, OLS, and a robust matrix of the variance of the estimator, which does not alter OLS estimates but makes them robust against heteroscedasticity and/or autocorrelation.

Furthermore, non-normality of residuals do not pose a threat to the consistency of OLS estimates. Even in the absence of normality, OLS would still be BLUE according to the Gauss-Markov theorem (Hill et al., 2011). Normality is only desirable as it implies that OLS is efficient also against non-linear estimators (Hayashi, 2011). In our application, the use of non-linear estimator is risky: these are typically based on maximization routines that have to be led to convergence. Even when a non-linear estimator has an analytical solution (for example, Maximum Likelihood Estimation (MLE) or Generalized Method of Moments (GMM)), in practice this solution has to be reached numerically. In our application, where we estimate several hundreds of models, it is difficult to control for convergence and for the presence of multiple maxima in each optimization. Moreover, efficiency of estimates of non-linear estimators (for example of MLE) can only be guaranteed asymptotically while in small samples other estimators may have greater concentration around the true parameter value (Pfanzagl, 1994). In this context, the use of OLS remains a safer choice.

In terms of influential observations, we present information on possible leverage issues. Leverage is a measure of how far away the independent variable values of an observation are from those of other observations. High-leverage points have the potential of being influential points, but they are not necessarily such (Cardinali, 2013). Leverage is based on the hat matrix $H = X(X^T X)^{-1} X^T$, where X is the matrix of explanatory variables used in a model. The diagonal elements of the hat matrix, h_{ii} , can be defined as a weighted distance between observation x_i to the mean of all x_i 's. A rule of thumb to identify leverage points is to identify the points whose leverage value exceeds two/three times the mean average leverage value. We provide statistics for both cases. In the case of the first threshold (two times the mean average leverage value), the average share of influential observations for models is 5.7%. In the case of the second threshold (3 times the average leverage value), the average share of influential observations for all models is 2.4%.

We are left in the dark as to how many farms lie behind the yield gap estimates.

>>> Reply

We have further clarified this within the text:

L106 ff: "The regional, crop-specific econometric models have been provided with farm-level data points ranging from as few as 36 to as many as 20'222 for regions and crops categories with extensive observations, with each observation representing a distinct farm."

Additionally, and most importantly, the Excel file 'input_npc_yield_gap_data.xlsx,' documents under columns F the number of total farms used and under columns G the number of organic farms used for each yield gap estimation.

(2) Including some form of interaction between techno-economic orientation and typology (organic/conventional) could be good because one could expect that yields are more similar in mixed farming, for instance, than in specialized crop farms. If organic farms only cover certain farm types I would suggest to restrict the input data to comparable farms rather than rely too much on the covariates to fix this. Given the large size of the FADN regions, I suspect that further grouping farms by techno-economic orientation (as an additional random effect) using that very same FADN data would help getting better estimates of the yield gaps which are central to this paper.

>>> Reply

Thank you for your comment, which fully aligns with our perspective. FADN provides a variable (TF8) that categorizes farms into eight techno-economic orientations: field crops, horticulture, wine, other permanent crops, milk, other grazing livestock, granivores, and mixed farms (see table below). Following your suggestion, we have incorporated an interaction between techno-economic orientation and the organic status of farms. However, instead of using random effects (fixed-effects at TF8 level were already included in the estimations), we opted for estimating a separate yield equation for each farm specialisation, crop, and region. This choice allows a fully flexible crop-region-specialization yield gap estimation. In the models we still control for a more detailed level of specialization (TF14) to capture additional yield differences attributed to higher levels of farm specialisations (for details on TF8 and TF14 classifications see table below). The use of TF8 instead of TF14 was strategic to maximize the statistical power of coefficient tests. After estimating the yield gaps for each crop-region-specialisation combinations, we obtain, regional yield gaps by taking the median of the yield gaps of the specialisations in each region. This approach, which is similar to that of Barreiro-Hurle et al. (2021), streamlines the computationally model estimation and provides a more flexible way to obtain specialisation-specific yield gaps. Importantly, this approach permits all coefficients of the models to vary across different specializations, ensuring the highest degree of flexibility.

TF14	TF8
15. Specialist COP	
16. Specialist other fieldcrops	1. Fieldcrops
61. Mixed cropping	
20. Specialist horticulture	2. Specialist horticulture
35. Specialist wine	3. Specialist wine
36. Specialist orchards - fruits	
37. Specialist olives	4. Other permanent crops
38. Permanent crops combined	
45. Specialist milk	5. Milk
48. Specialist sheep and goats	
49. Specialist cattle	6. Other grazing livestock
50. Specialist granivores	7. Granivores
70. Mixed livestock	
80. Mixed crop and livestock	8. Mixed

Furthermore, in regions where we lacked sufficient organic observations to segment the analysis at the TF8 level, we estimated the yield gap using all typologies collectively (previous approach). This approach resulted in 77% of yield gaps derived from segregated TF8 observations, while 23% of estimations stemmed from utilizing observations across all typologies for a single region and crop.

For clarification, we have incorporated the following information in the Methods section:

L371 ff: " Using FADN data, we confine the estimation of yield gaps to each FADN region and, whenever possible, to each farm typology to ensure both representative data and limit the heterogeneity of the operating environment of farms. In regions with over 16 reported organic farms per farm typology and crop, we calculate within each region yield gaps for each farm typology and crop. These crop-specific yield gap estimates per typology are then averaged regionally to obtain a single estimate for each crop in that region. In regions with insufficient organic observations to segment the analysis by farm typology, we derive a regional yield gap by aggregating different farm types together. Additionally, crop-region combinations with fewer than 16 organic farms across all farm typologies were excluded entirely from the analysis. Using this approach, 77% of yield gaps were derived from farm typology-specific estimations, while 23% were obtained from estimations across farm typologies. "

Accordingly, we also have altered the short description in the results section to:

L98 ff: "Using data from 2010 to 2017, yield gaps are estimated for each FADN region of the EU and by farm typology, provided that a minimum number of organic farms per farm typology and crop operate in a region."

(3) There is inclusion of mineral fertilizers in the model, but not of other types of fertilizers that are used in organic farming, i.e. manure, slurry, and the input of nitrogen through grass-clover leys. This could potentially cause a bias in the estimates of yield gaps. I expect that organic farmers may often be close to matching fertilization levels of their conventional counterparts.

>>> Reply

We agree that a possible bias, even though small, could exist in this case. To address this issue, we have looked into the data again and resolved the issue of manure, which is a significant source of fertilizers for crop production. EUROSTAT (2018) reports that mineral fertilisers accounted for 45% of nitrogen input in the EU in 2014, while manure contributed 38%. However, there is no information provided on how manure is split between conventional and organic farming (the latter representing only 9.9% of the EU UAA). Furthermore, the use of nitrogen from animal manure has been limited to 170 kg N/ha by EU regulations (Köninger et al. 2021). Many crops need more than 170 kg N/ha and are permitted by law to receive this. This is then supplemented with synthetic nitrogen fertiliser in conventional farming, which is not allowed in organic farming. However, organic farming has a wider, less intensive crop rotation, and also legumes are often used to bind nitrogen, reducing requirements for additional nitrogen. Nitrogen bound by legumes is not counted in the balance as it is calculated based on the supply of manure, and as a result, the calculated nitrogen efficiency seems better. So indeed, if an organic farm optimises its choice of crop, crop order, fertilisation, and use of green fertilisers/catch crops, then the nitrogen efficiency does not need to be lower than on a conventional farm. Even though some farms may achieve nitrogen efficiency, several others might not, in particular for the most nitrogen demanding ones, such as cereal farms for instance.

Coming back to the point on how we could capture this in the model, unfortunately FADN data on manure are mostly non-existing and if available (only happens in very rare occasion) not reliable. Data on grass-clover coverage is also not available. As a consequence, we addressed the possible bias occurring from manure introducing the following two variables to account for non-synthetic fertilizers use: 1) fertilisers and soil improvers (in EUR) per hectare of utilised agricultural area (UAA), and 2) the total number of livestock units divided by the hectares of UAA.

The variable “fertilisers and soil improvers” represents the cost of all purchased fertilisers and soil improvers (e.g. lime) including compost, peat and manure (excluding manure produced on the holding). This indicator divided by the total UAA is thus a proxy for the use of purchased fertilisers on the farm. Please note that this variable was already included in the previous estimation, however we did not correctly mention its definition. We corrected this in the revised manuscript and reemphasize here its use and definition. Moreover, it is important to emphasize that the variable fertiliser and soil improvers excludes manure produced on the holding. As there is no information regarding manure produced on the holding in the FADN survey, we assume a proportional relationship between manure use and the number of livestock units on the farm. Total livestock units are the sum of equines, cattle, sheep, goats, pigs, and poultry (annual average number), converted into a standardised measure, i.e., livestock units. Livestock units are determined by multiplying each animal category by a conversion coefficient representing the nutritional or feed requirement of each type of animal². The reference unit used for the calculation of livestock units (=1 LSU) is the grazing equivalent of one adult dairy cow producing 3,000 kg of milk annually, without additional concentrated feedstuffs. As feed requirements are correlated with manure production, the total number of livestock units per hectare proxies the amount of available manure on the holding that could be used as fertilizer. Additional information on FADN variables can be found at European Commission (2020a).

Also, it would be important to ensure that your explanatory variables are not too strongly correlated with each other so as to hamper the estimation of the effects (beta1). In fact, it is probably sufficient that the X1 variables are not too strongly correlated with the organic variable since that it is the coefficient for that variable that is used later on.

>>> Reply

Regarding the potential correlation between the organic dummy and other explanatory variables, we acknowledge that some degree of correlation is expected due to inherent associations among farming practices. However, collinearity in this context does not compromise the consistency of model estimates. While high collinearity might reduce the efficiency of estimators (Hill et al., 2011), given that we have not emphasised the significance of model estimates, the presence of some collinearity is not an important concern. Nonetheless, to address collinearity, we conducted a variance inflation factor (VIF) test, where the organic status of the farm is regressed against all other explanatory variables present in the yield model (Gareth et al., 2017). The VIF test quantifies the degree by which a dependent variable, in our case the organic status of the farm, can be predicted by the remaining explanatory variables. The test is implemented using the following formula:

² [https://ec.europa.eu/eurostat/statistics-explained/index.php?title=Glossary:Livestock_unit_\(LSU\)](https://ec.europa.eu/eurostat/statistics-explained/index.php?title=Glossary:Livestock_unit_(LSU)) accessed 3 January 2023.

$$VIF_{\beta_i} = \frac{1}{1 - R_i^2}$$

Where VIF_{β_i} represents the VIF for the coefficient β associated to the organic dummy in model i and R_i^2 is the coefficient of determination to that model. Typically ad hoc thresholds to define high multicollinearity are set at 10 or 5. By choosing a threshold of 10, 97% of the models do not present a high degree of multicollinearity. When reducing the threshold to 5, this percentage decreases to 94%. Overall, the analysis indicates that there are very few cases in which multicollinearity poses a threat to the inference on the coefficients, reinforcing the robustness of our results.

(5) There is cleaning of outliers going on in the code, that needs to be explained and motivated in the paper.

>>> Reply

We have added information on our data cleaning procedures in the Method section:

L367 ff: *“Before performing the yield gap estimations, we carry out data cleaning based on the examination of yield data. For each Member State, we analyse the yield distribution for each crop. The data cleaning is performed by removing the extreme 1% from both tails of each distribution, i.e., the most extreme 2% of the observations are removed. These data points are identified as outliers and often deviate from biophysical plausible yield values.”*

We do not perform any other data cleaning procedure as we would expect that several organic farms present apparently limiting values of, for example, input data. Furthermore, cleaning only the data from extreme yield values preserves the already limited number of organic farms while excluding those values that are most likely outliers.

(6) The final data only contains a subset of the regions

>>> Reply

Yes, we only estimate yield gaps for regions in which at least 16 crop-specific organic observations can be found. This leads to a subset of regions, which are the ones shown in the Excel file `input_npc_yield_gap_data.xlsx`.

* on the NPC mixed model:

Model diagnostics are also needed here. The residuals are not independent of spatial location... There is a significant effect of the relative size of the ratio of evidence for organic to evidence for conventional on the estimate, as well as of the biogeographic variable. Together this reduces the estimate for the standardized LF_NPC by almost half, from 6.1 to 3.3. It is still significant (P=0.02). R code for that:

```
explore_lmer = lmer(estimate ~ l(nobs_organic/nobs)+biogeo+median_biogeo_norm + (1 | crop),
data = npc_clean)
car::Anova(explore_lmer)
```

To complement with further model diagnostics also for the mixed model, we display the Tukey-Anscombe plot, where we plot the residuals (y-axis) against the yield gaps i.e. the fitted values (x-axis). The graph shows that the residuals are constant and do not change with the fitted values, hence

confirming the assumption of homoscedasticity. Furthermore, the Tukey-Anscombe plot shows that the errors are normally distributed around 0, fulfilling the assumption of normality. Latter is further confirmed when plotting a histogram of the residuals:

Furthermore, thank you for providing above code. We agree and see the same when executing the code you have provided, except for the relative size between organic and total observations. With the implementation of the new model, which includes updated LF-NPC and yield gap estimation, we found that the significance of the variable capturing the ratio between organic and conventional observations exceeded the 0.1 threshold. Nevertheless, we still investigated the meaning of including this variable (since the threshold was only slightly exceeded). We found that when the biogeographic regions are not included as dummy variables (as per the argument below), the variable capturing the ratio does not change the estimation of the LF-NPC coefficient by much, in fact it changes from 4.36 to 4.1. Furthermore, within the context of our research question, which focuses on LF-NPC versus yield gaps, we found that this variable adds little substantive meaning. Nevertheless, this observation could be added in the Discussion section, where we mention the limitations of the yield gap estimation (L262 ff). In summary, within the mixed model we argue to exclude this variable, as it enhances the model's interpretability without significantly altering the LF-NPC coefficient, the main focus point of our paper.

The context is different when including biogeographic regions into the mixed model, where we still see a significant threshold below 0.1. As shown in the accompanying boxplot, the distribution of yield gaps varies across biogeographic regions. Notably, the Mediterranean region exhibits the lowest average yield gap. This is partly attributed to the diverse crop mix within each biogeographic region, which contributes to the observed differences in yield gap distributions across crop groups.

While we could consider adding the biogeographic region as dummy, we believe that the available data may not be sufficient to support this level of granularity. The estimated yield gap dataset often contains sparse observations for a given crop within a particular biogeographic region. For example, durum wheat has only one yield gap observation in the biogeographic region "Alpine", and only three in the continental area, while the remaining 16 observations on yield gaps for durum wheat are recorded for the Mediterranean region. Since two regions ("Mediterranean" and "Continental") hold most observations, including biogeographic regions as dummies might be misleading. Further research with a more comprehensive dataset would be required to fully explore the interaction between these factors.

Furthermore, and most importantly, we intentionally normalized LF-NPC by biogeographic region to ensure comparability across regions belonging to different biogeographic areas. Therefore, we suggest that it is more accurate to pull the observations across different biogeographic regions together when evaluating the impact of LF-NPC on yield gaps without adding the biogeographic regions as dummies, which further offers the advantage of a larger sample size, and more degrees of freedom.

On CAPRI modelling:

There is insufficient detail to allow assessment of the way in which yield gaps estimates were introduced into the model. If I am not mistaken, the following is literally the only information available in the paper "In this study, we project all EU regions to reduce their pesticides use to levels observed under organic farming by 2030, and adopt the estimated yield gaps, taking into account each region's NPC potential and the pesticides reduction. The latter is implemented by changing yields and pesticides cost under the regional agricultural supply modules of CAPRI".

L370 which CAPRI supported stable release (STAR) was used?

>>> Reply

We added a paragraph in the Supplementary Information (L198 ff) on how we introduce the yield shocks (exogenously) and how the model endogenously reacts to these shocks in the scenario with and without market feedback.

L198 ff: *“When using the model without market feedback, the exogenous yield changes introduced trigger endogenous responses depending on the relative profitability between crops and price elasticities 10. When running the CAPRI model with the market feedback activated, the endogenous reaction depends on an iterative process triggered by supply and price changes in the global and local markets, following the micro economic theory between supply and demand until an equilibrium is reached between supply and demand and the market balance is achieved.”*

As regards which CAPRI version was used, both scenarios were run under the latest available CAPRI version at the time of the analysis (i.e. we use the latest trunk version of CAPRI). This allowed us to make use of an updated database and use 2017 as the base year. We did not use a stable release version, because that would include neither the updated database nor base year 2017. We put this information in the Supplementary Information (line 195 ff).

Other comments

Abstract: The statement “6.1% increase in agricultural income” in the abstract (L14) is not correct, as this is the reduction in the yield gap observed (L134).

>>> Reply

We have corrected this in the revised manuscript, also responding to a suggestion made earlier.

L31 replace “such as” with “including amongst others”, as it is hard to have a complete list here (you missed bats and spiders, for instance)

>>> Reply

We have replaced the expression. Thank you.

L42-44 but see Karp et al. (cited in the study) which finds inconsistent effects.

>>> Reply

Recognizing that we discuss the inconsistency of the results further below, we opted to maintain the original structure while making sure Karp et al. is cited and substituting "inconsistent" for "inconclusive."

L161 and L163 Provence-Alpes-Côte d'Azur

>>> Reply

Corrected accordingly.

L183 it is not species richness but abundance (Rega et al. 2018 and papers cited therein)

>>> Reply

Corrected accordingly.

L309-311 how can sub-regional data be accessed if we do not know where the farms are within the FADN region?

>>> Reply

We do not include sub-regional dummies in plural, but only one which is the NUTS3 region a farm belongs to, which is provided under FADN. Nevertheless, as explained earlier, we cannot model at that level given the scarcity of farms reporting at NUTS3.

L328 mention that the map is only considering agricultural areas.

>>> Reply

This is addressed, as we now (given the new variant of LF-NPC used) provide much more detail in the Supplementary Information section one on the LF-NPC scores and what areas we take into account.

Figure 4 - when "region" is used, mention which region it is (there are multiple ones in play in this study).

>>> Reply

Thank you for this comment, we have added which region we refer to in Figure 4 and Figure 5.

Additional references, mentioned in the replies to reviewer 2:

Barreiro-Hurle, J., Bogonos, M., Himics, M., Hristov, J., Pérez-Domiguez, I., Sahoo, A., Salputra, G., Weiss, F., Baldoni, E., Elleby, C. (2021). Modelling environmental and climate ambition in the agricultural sector with the CAPRI model. Exploring the potential effects of selected Farm to Fork and Biodiversity strategies targets in the framework of the 2030 Climate targets and the post 2020 Common Agricultural Policy, EUR 30317 EN, Publications Office of the European Union, Luxembourg, 2021, ISBN 978-92-76-20889-1, doi:10.2760/98160, JRC121368.

Bianchi, F. J. J. A., Booij, C. J. H., & Tschardtke, T. (2006). Sustainable pest regulation in agricultural landscapes: A review on landscape composition, biodiversity and natural pest control. In Proceedings of the Royal Society B: Biological Sciences (Vol. 273, Issue 1595). <https://doi.org/10.1098/rspb.2006.3530>

CO-FREE Consortium (2016). "Final Report Summary - CO-FREE (Innovative strategies for copper-free low input and organic farming systems)." European Commission. 2016. <https://cordis.europa.eu/project/id/289497/reporting>.

Cardinali, C. (2013). Data Assimilation. Observation influence diagnostic of a data assimilation system. European Centre for Medium-Range Weather Forecasts Research Department. Series: ECMWF Lecture Notes. At: <https://www.ecmwf.int/sites/default/files/elibrary/2013/16938-observation-influence-diagnostic-data-assimilation-system.pdf>.

Cortignani, R., and Dono, G. (2020). Greening and legume-supported crop rotations: An impact assessment of Italian arable farms. Science of the Total Environment, Volume 734, 10 September 2020, 139464, <https://doi.org/10.1016/j.scitotenv.2020.139464>.

De Ponti, T., Rijk, B., & Van Ittersum, M. K. (2012). The crop yield gap between organic and conventional agriculture. Agricultural Systems, 108. <https://doi.org/10.1016/j.agsy.2011.12.004>

European Commission (2020a). Farm Return Definitions. RI/CC 1680 v6, Brussels, May 2020.

European Commission (2020b). Farm Accounting Data Network. An A to Z Methodology. Version 31/10/2020. Available at: <https://circabc.europa.eu/ui/group/befb6055-ab0c-4305-84fe-0c80c1c0553d/library/1df3a121-11ee-40c3-a991-70a5f3cdd9d7/details>.

European Parliament and Council of the European Union (2018). "Regulation (EU) 2018/848 on organic production and labelling of organic products and repealing Council Regulation (EC) No 834/2007."

Official Journal of the European Union L150 (June 14): 1-92. <https://eur-lex.europa.eu/legal-content/EN/TXT/?uri=CELEX:32018R0848>.

EUROSTAT (2018). Agri-environmental indicator - gross nitrogen balance. [https://ec.europa.eu/eurostat/statistics-explained/index.php?title=Archive:Agri-environmental indicator - gross nitrogen balance](https://ec.europa.eu/eurostat/statistics-explained/index.php?title=Archive:Agri-environmental_indicator_-_gross_nitrogen_balance)

Fogliatto, S., Andres, A., Concenço, G., Vidotto, F., & Knezevic, S. (2023). Editorial: Weed management in organic agriculture. *Frontiers in Agronomy*, 4. <https://doi.org/10.3389/fagro.2022.1116519>

Gareth, J., Witten, D., Hastie, T., Tibshirani, R. (2017). *An Introduction to Statistical Learning* (8th ed.). Springer Science+Business Media New York. ISBN 978-1-4614-7138-7.

Hayashi, F. (2011). *Econometrics*. Princeton University Press, ISBN 9781400823833.

Hill, R.C., Griffiths, W.E., Lim, G.C. (2011). *Principles of Econometrics*, Fourth Edition. John Wiley & Sons, Inc. ISBN 978-0-470-62673-3.

IFOAM (2018). Strategy for the minimisation of copper in organic farming in Europe (IFOAM EU Group), May 2018. https://www.organicseurope.bio/content/uploads/2020/10/ifoam_eu_copper_minimisation_in_organic_farming_may2018_0.pdf?dd

Knapp, S., & van der Heijden, M. G. A. (2018). A global meta-analysis of yield stability in organic and conservation agriculture. *Nature Communications*, 9(1). <https://doi.org/10.1038/s41467-018-05956-1>

Königer, J., Lugato, E., Panagos, P., Kochupillai, M., Orgiazzi, A., & Briones, M. J. I. (2021). Manure management and soil biodiversity: Towards more sustainable food systems in the EU. In *Agricultural Systems* (Vol. 194). <https://doi.org/10.1016/j.agsy.2021.103251>

Kremmydas, D., Ciaian, P., Baldoni, E. (2024). Modeling conversion to organic agriculture with an EU-wide farm model. *Bio-based and applied economics*, 10.36253/bae-13925.

Letourneau, D., & Bruggen, A. van. (2017). Crop protection in organic agriculture. In *Organic agriculture: a global perspective*. <https://doi.org/10.1079/9781845931698.0093>

Litterick, A. M., Watson, C. A., & Atkinson, D. (2002). Crop protection in organic agriculture – a simple matter? *Proceedings of the UK Organic Research 2002 Conference: Research in Context*. 26-28th March 2002 Aberystwyth.

Lundkvist, A., & Verwijst, T. (2011). Weed Biology and Weed Management in Organic Farming. In *Research in Organic Farming*. <https://doi.org/10.5772/31757>

Morrone, J. J. (2018). The spectre of biogeographical regionalization. *Journal of Biogeography*, 45(2), 282–288. <https://doi.org/https://doi.org/10.1111/jbi.13135>

Oerke, E. C. (2006). Crop losses to pests. In *Journal of Agricultural Science* (Vol. 144, Issue 1). <https://doi.org/10.1017/S0021859605005708>

Paracchini, M.-L., & Britz, W. (2010). Quantifying effects of changed farm practise on biodiversity in policy impact assessment – an application of CAPRI-Spat.

Petsakos, A., Ciaian, P., Espinosa, M., Perni, A., and Kremmydas, D. (2023). Farm-level impacts of the CAP post-2020 reform: A scenario-based analysis. *Applied Economic Perspectives and Policy*, Volume 45, Issue 2, <https://doi.org/10.1002/aep.13257>.

Pfanzagl, J. (1994). *Parametric Statistical Theory*. Published by De Gruyter, <https://doi.org/10.1515/9783110889765>.

Ponisio, L. C., M'gonigle, L. K., Mace, K. C., Palomino, J., Valpine, P. De, & Kremen, C. (2015). Diversification practices reduce organic to conventional yield gap. *Proceedings of the Royal Society B: Biological Sciences*, 282(1799). <https://doi.org/10.1098/rspb.2014.1396>

Royston, J. P. (1982). Algorithm AS 181: The W Test for Normality. *Applied Statistics*, Vol. 31, No. 2. (1982), pp. 176-180.

Royston, J. P. (1995). Remark AS R94: A Remark on Algorithm AS 181: The W-test for Normality. *Journal of the Royal Statistical Society. Series C (Applied Statistics)*, Vol. 44, No. 4 (1995), pp. 547-551 (5 pages), Published By: Oxford University Press.

Savary, S., Willocquet, L., Pethybridge, S. J., Esker, P., McRoberts, N., & Nelson, A. (2019). The global burden of pathogens and pests on major food crops. *Nature Ecology and Evolution*, 3(3). <https://doi.org/10.1038/s41559-018-0793-y>

Seufert, V., & Ramankutty, N. (2017). Many shades of gray—the context-dependent performance of organic agriculture. *Science Advances*, 3(3). <https://doi.org/10.1126/sciadv.1602638>

Seufert, V., Ramankutty, N., & Foley, J. A. (2012). Comparing the yields of organic and conventional agriculture. In *Nature* (Vol. 485, Issue 7397). <https://doi.org/10.1038/nature11069>

Tamm, L., Thuerig, B., Apostolov, S., Blogg, H., Borgo, E., Corneo, P. E., Fittje, S., de Palma, M., Donko, A., Experton, C., Marín, É. A., Pérez, Á. M., Pertot, I., Rasmussen, A., Steinshamn, H., Vetemaa, A., Willer, H., & Herforth-Rahmé, J. (2022). Use of Copper-Based Fungicides in Organic Agriculture in Twelve European Countries. *Agronomy*, 12(3). <https://doi.org/10.3390/agronomy12030673>

Wilk, M.B., and Gnanadesikan, R. (1968), Probability plotting methods for the analysis of data. *Biometrika*, Biometrika Trust, 55 (1): 1–17.

Reviewer #3 (Remarks to the Author):

This study aims to quantify the benefits of landscape features that promote natural pest control (LF-NPC) for multiple crops across Europe. The authors estimate the yield gaps between organic production and conventional production and then use a map of LF-NPC potential in Europe to analyze the relationship between the yield gap and LF-NPC potential. They also use a model to predict the changes in yield gap associated with a 50% reduction in pesticides, which is the target reduction under the Biodiversity Framework for the year 2030. Finally, the study estimates how LF-NPC would affect agriculture income under that scenario. The results suggest that areas with high LF-NPC potential experience lower productivity losses due to pesticide reduction. I find the approach presented in this paper very interesting, and the results are significant. The concepts are well-explained, and the paper is written very well. The major conclusions align with the results. However, I have a few points that I would like the authors to address to make this paper stronger.

1) The study does not account for the potential changes in biodiversity (and in turn on natural pest control) that could result from a decrease in pesticide use. Their model assumes that any fluctuations in biodiversity and natural pest control arise solely from landscape features, and not from the long-standing history of pesticide use in the area. As a result, the model appears to be static in this regard. It would be interesting to incorporate dynamic features that demonstrate how reducing pesticide usage could lead to an increase in biodiversity and contribute to NPC. It's possible that the impact of pesticide use on biodiversity is even greater than that of landscape features in highly intensive farming regions. Regardless of whether the authors choose to include this dynamic component in their model, they should discuss the potential impact of decreasing pesticide use on biodiversity.

>>> Reply

Thank you for your comment regarding the effects of pesticide use on biodiversity and its consequent impact on NPC. Indeed, it is correct that the simulation in this study assesses a static potential of landscape features to support NPC, without directly accounting for changes in biodiversity due to varying pesticide intensities. This limitation arises from two main factors.

Firstly, the current capabilities of the CAPRI model do not allow for dynamically simulating biodiversity changes over time. Hence, the model is unable to quantify the potential impact of changes in biodiversity resulting from reduced pesticide use. Therefore, the study relies on a static representation of LF-NPC in the EU and projects this into the future.

Secondly, and this further highlights the reason why it has not yet been incorporated in this study: We are limited by the lack of sufficient studies quantifying the enhancement of biodiversity, or more specifically, NPC when fewer pesticides are applied (and vice-versa). A single coefficient that we could use in our model to evaluate biodiversity changes in relation to pesticide intensity is not available as far as we know. As a result, we refrained from attempting to use a specific numerical value. However, we acknowledge the potential, and for future (different) analysis we do not discard the possibility of doing so, especially given the growing data availability and research efforts with respect to changes in biodiversity. A possible pathway could be by making use of results from process-explicit models, yet to be developed, that could integrate the generalizable mechanisms responsible for the distributions of organisms, communities, and ecosystems in space and time.

Finally, and most importantly, we also see benefits in focusing on a static biodiversity status, as it allows us to isolate the benefits of LF-NPC on yield gaps from other possible impacts, such as pesticides

use. While enhancing NPC through reduced pesticide is another important research area, it is not the focus point of this paper. To that purpose and as suggested by the reviewer we chose to discuss this additional perspective in the manuscript. We have added the following in the Discussion section:

L311 ff: "Along this line of thought, it is important to also acknowledge that pesticide use intensity, reduced to organic farming levels under this study, can have considerable impact on biodiversity dynamics (Sigmund et al., 2023) and, consequently, can affect natural enemies (Passos et al., 2022). However, within the CAPRI framework, there is currently no module that incorporates a coefficient to assess the impact of pesticide reduction on biodiversity, as finding such a universal measure is challenging due to its context-specific nature (Ricci et al., 2019). Ideally, future research could look towards employing process-explicit models that better account for the mechanisms influencing the distribution of organisms and communities, and how pesticides affect them at various scales. Integrating such modules that dynamically capture the complex interplay between pesticides, biodiversity, and NPC into CAPRI would significantly enhance its capabilities."

2) Although the paper focuses on examining the impact of landscape-level features on biodiversity and NPC, it's crucial to consider the effect of local-level diversification as well. Intercropping, cover crops, rotations, silvopastoral systems, small-scale hedge rows, etc. are some of the examples of local-level diversification that can have a greater impact on increasing biodiversity and NPC than the landscape-level features. Therefore, the paper should also discuss the importance of local-level diversification and cite relevant literature examining its effects.

>>> Reply

Many thanks for your comment emphasizing the importance of local-level diversification in enhancing biodiversity and NPC. As you correctly point out, practices like intercropping, cover crops, crop rotations, silvopastoral systems, and small-scale hedgerows are pivotal in fostering biodiversity and supporting NPC, which in turn can influence yield gaps between organic and conventional farming. We have now included the following sentences in the Discussion section, which addresses this aspect:

L321 ff: "... local diversification strategies such as intercropping, crop rotations, cover crops, and small-scale hedgerows (Jones et al., 2023), although context-dependent, can offer farmers practical and timely solutions to pest management challenges (Rosa-Schleich et al., 2019). These practices are known for their potential to enhance the presence and effectiveness of beneficial insects and other natural enemies in agricultural ecosystems (Smith et al., 2020), without necessarily taken land away from agricultural production, which could be the case when incorporating landscape features into agricultural landscapes. Therefore, such strategies can play a major role in reducing yield gaps in organic systems compared to conventional farming (Ponisio et al., 2015). "

3) The model fails to consider an important factor which is the land that may need to be taken away from production to incorporate landscape-level features. This aspect is closely linked to the previous one where diversifying within plots through intercropping and other techniques can increase NPC without sacrificing the yield or losing land that is currently in use.

>>> Reply

We agree that this is an important consideration. To address this point, we have incorporated the issue of land conversion away from agricultural land into landscape-features into the discussion alongside to our response to your previous comment.

L323 ff: *"These practices are known for their potential to enhance the presence and effectiveness of beneficial insects and other natural enemies in agricultural ecosystems (Smith et al., 2020), **without necessarily taking land away from agricultural production.**"*

Furthermore, in the following lines of the manuscript, we also make reference to the fact that we do not analyse the costs associated to landscape feature in agricultural landscapes:

L203 ff: *"Furthermore, although we explore the potential positive feedback loop of an enhanced landscape design supporting NPC, we only accounted for the costs related to reduced pesticides use and did not explore the costs associated with redesigning the agricultural landscape to increase LF-NPC potential. Further investigation is needed to estimate these costs, which may involve leveraging diverse data sources."*

Minor comments:

1) Please provide clarification on the specific costs associated with reduced pesticide use as mentioned in lines 196-198.

>>> Reply

To model the reduction in pesticide use to levels observed under organic farming in 2030, we considered both yield changes and the reduction in costs associated with lower or no pesticide use. Our estimation was based on the FADN data for the variable 'crop protection products', which represents the reduction in costs of pesticides used in fully organic farms compared to conventional farms. Even if pesticides are reduced to zero use, farmers still incur other crop protection costs. Therefore, we assumed an 80% cost reduction. Although we cannot share the FADN data for confidentiality reasons, another document from the European Commission, based on the same data, indicates that the cost reduction typically ranges between 75 to 100%, depending on the crop and region (European Commission, 2023). As such, assuming an 80% reduction is reasonable for the purposes of this analysis.

Additionally, please elaborate further on how the extent of the y-axis in relation to the data presented explains the point made in line 214. Thank you.

>>> Reply

In Figure 3, we see most of the points show a negative income change (y-axis) as a result of declining yields inserted into the CAPRI model. We think making reference to the y-axis at this section of the manuscript does not add value. We have therefore opted for taking the parenthesis "(see extent of y-axis Figure 3)" out.

2) Figure 1 should include significance levels in addition to correlation coefficients.

>>> Reply

We understand that this comment refers to the current Figure 2. Figure 1 presents the distribution of yield gap estimates for each crop. In contrast, Figure 2 illustrates the correlation coefficients between LF-NPC and the yield gaps. We have decided to modify the display by positioning the correlation coefficients in the upper right corner of each chart rather than in the title as was previously done. We

find, that this change allows for a more intuitive visual analysis of the data and makes the charts less busy. Following this strategy, we have opted for providing the requested information in the Supplementary Information, which provides a detailed explanation of various LF-NPC variants and their significance values with respect to explaining yield gaps per crop. See section one in the Supplementary Information. We hope that this is sufficient, alternatively, we can keep trying to put them on the charts, without making them look too busy.

3) Figure 3 should include the regression coefficient and significance value.

>>> Reply

We agree and added the regression coefficient, p-value and R-square in the lower right corner of the chart.

Additional references, mentioned in the replies to reviewer 3:

European Commission. A Decade of Organic Growth Organic Farming in the EU N°20 Agricultural Market Brief.; 2023. https://agriculture.ec.europa.eu/system/files/2023-04/agri-market-brief-20-organic-farming-eu_en.pdf

Ponisio, L. C., M'gonigle, L. K., Mace, K. C., Palomino, J., Valpine, P. De, & Kremen, C. (2015). Diversification practices reduce organic to conventional yield gap. *Proceedings of the Royal Society B: Biological Sciences*, 282(1799). <https://doi.org/10.1098/rspb.2014.1396>

Jones, S. K., Sánchez, A. C., Beillouin, D., Juventia, S. D., Mosnier, A., Remans, R., & Estrada Carmona, N. (2023). Achieving win-win outcomes for biodiversity and yield through diversified farming. *Basic and Applied Ecology*, 67. <https://doi.org/10.1016/j.baae.2022.12.005>

Passos, L. C., Ricupero, M., Gugliuzzo, A., Soares, M. A., Desneux, N., Carvalho, G. A., Zappalà, L., & Biondi, A. (2022). Does the dose make the poison? Neurotoxic insecticides impair predator orientation and reproduction even at low concentrations. *Pest Management Science*, 78(4). <https://doi.org/10.1002/ps.6789>

Ricci, B., Lavigne, C., Alignier, A., Aviron, S., Biju-Duval, L., Bouvier, J. C., Choisis, J. P., Franck, P., Joannon, A., Ladet, S., Mezerette, F., Plantegenest, M., Savary, G., Thomas, C., Vialatte, A., & Petit, S. (2019). Local pesticide use intensity conditions landscape effects on biological pest control. *Proceedings of the Royal Society B: Biological Sciences*, 286(1904). <https://doi.org/10.1098/rspb.2018.2898>

Rosa-Schleich, J., Loos, J., Mußhoff, O., & Tschardtke, T. (2019). Ecological-economic trade-offs of Diversified Farming Systems – A review. In *Ecological Economics* (Vol. 160). <https://doi.org/10.1016/j.ecolecon.2019.03.002>

Sigmund, G., Ågerstrand, M., Antonelli, A., Backhaus, T., Brodin, T., Diamond, M. L., Erdelen, W. R., Evers, D. C., Hofmann, T., Hueffer, T., Lai, A., Torres, J. P. M., Mueller, L., Perrigo, A. L., Rillig, M. C., Schaeffer, A., Scheringer, M., Schirmer, K., Tlili, A., ... Groh, K. J. (2023). Addressing chemical pollution in biodiversity research. *Global Change Biology*, 29(12). <https://doi.org/10.1111/gcb.16689>

Smith, O. M., Cohen, A. L., Reganold, J. P., Jones, M. S., Orpet, R. J., Taylor, J. M., Thurman, J. H., Cornell, K. A., Olsson, R. L., Ge, Y., Kennedy, C. M., & Crowder, D. W. (2020). Landscape context affects the sustainability of organic farming systems. *Proceedings of the National Academy of Sciences of the United States of America*, 117(6). <https://doi.org/10.1073/pnas.1906909117>

Reviewers' Comments:

Reviewer #1:

Remarks to the Author:

The authors have done a really thorough and impressive job responding to my previous concerns and they have adjusted their models and discussions accordingly. I have no further comments. Congratulations to an important paper!

Reviewer #2:

Remarks to the Author:

The revision has been very thorough, and the changes outline in detail in the rebuttal letter, which has been very useful. I see all my comments have been addressed satisfactorily.

Reviewer #3:

Remarks to the Author:

I am satisfied with the revisions made. think that the manuscript is stronger, and better documented, and the results are better supported.